# High-voltage and dendrite-free zinc-iodine flow battery

Caixing Wang ●[1] ✉, Guoyuan Gao[1], Yaqiong Su ●[2], Ju Xie[1], Dunyong He[1], Xuemei Wang[1], Yanrong Wang ●[1] ✉ & Yonggang Wang ●[3] ✉

$Zn\text{-}I_2$ flow batteries, with a standard voltage of 1.29 V based on the redox potential gap between the $Zn^{2+}$-negolyte (−0.76 vs. SHE) and $I_2$-posolyte (0.53 vs. SHE), are gaining attention for their safety, sustainability, and environmental-friendliness. However, the significant growth of Zn dendrites and the formation of dead Zn generally prevent them from being cycled at high current density (>80 mA cm$^{-2}$). In addition, the crossover of $Zn^{2+}$ across cation-exchange-membrane also limits their cycle stability. Herein, we propose a chelated $Zn(P_2O_7)_2^{6-}$ (donated as $Zn(PPi)_2^{6-}$) negolyte, which facilitates dendrite-free Zn plating and effectively prevents $Zn^{2+}$ crossover. Remarkably, the utilization of chelated $Zn(PPi)_2^{6-}$ as a negolyte shifts the $Zn^{2+}/Zn$ plating/stripping potential to −1.08 V (vs. SHE), increasing cell voltage to 1.61 V. Such high voltage $Zn\text{-}I_2$ flow battery shows a promising stability over 250 cycles at a high current density of 200 mA cm$^{-2}$, and a high power density up to 606.5 mW cm$^{-2}$.

Addressing the challenge posed by the growing global energy demand in conjunction with environmental concerns necessitates the development of sustainable, large-scale renewable energy sources such as solar and wind power[1,2]. However, the inherent variability in weather conditions and light intensity significantly impacts the reliability of photovoltaic and wind power generation, making them less than ideal for seamless grid integration[3]. In this context, aqueous redox flow batteries (ARFBs) have emerged as a promising solution[1,4]. These batteries offer the advantage of separating the energy storage medium from the reaction sites, effectively mitigating the intermittency associated with renewables. Moreover, ARFBs can decouple power and energy, all while meeting stringent safety requirements due to the features of excellent scalability, modular manufacturing, flexible design, as well as the non-flammability of aqueous electrolytes[5]. As an illustration, all-vanadium ARFBs are currently the most widely commercialized RFB system[6]. Nevertheless, their widespread adoption is still constrained by the substantial cost of vanadium[2,5]. Therefore, significant efforts have been dedicated to

exploring alternative ARFB technologies that are more economically viable[7–11].

In addition to the fully soluble ARFBs mentioned above, zinc-based flow batteries have also made great strides in scaled energy storage due to the inexpensive zinc electrolyte, which can now reach the MW/MWh level[12]. In recent years, $Zn\text{-}I_2$ flow batteries (ZIFBs) with a standard voltage of 1.29 V, derived from the redox potential difference between the $Zn^{2+}$-negolyte ($Zn^{2+}/Zn$ at −0.76 V vs. SHE) and the $I_2$-posolyte ($I_3^-/I^-$ at 0.53 V vs. SHE), have garnered interest due to their safety, sustainability, and eco-friendliness[13–15]. However, ZIFBs typically encounter issues such as undesirable Zn-dendrite growth and corrosion induced by proton, which not only compromise cycle stability but also restrict the achievable current density[16]. It is widely recognized that the growth of Zn-dendrites on the anode becomes increasingly severe at higher charging current densities (in mA cm$^{-2}$), significantly elevating the risk of short circuits[17]. During subsequent discharge cycles, the dissolution of Zn at the dendrite roots results in the formation of inactive or 'dead' Zn, consequently reducing the effective

---

[1]Institute of Innovation Materials and Energy, School of Chemistry and Chemical Engineering, Yangzhou University, Yangzhou, Jiangsu, China. [2]School of Chemistry, Xi'an Key Laboratory of Sustainable Energy Materials Chemistry, State Key Laboratory of Electrical Insulation and Power Equipment, Engineering Research Center of Energy Storage Materials and Devices of Ministry of Education, Xi'an Jiaotong University, Xi'an, China. [3]Department of Chemistry and Shanghai Key Laboratory of Molecular Catalysis and Innovative Materials, Institute of New Energy, iChEM (Collaborative Innovation Center of Chemistry for Energy Materials), Fudan University, Shanghai, China. ✉e-mail: caixingwang@yzu.edu.cn; yanrongwang@yzu.edu.cn; ygwang@fudan.edu.cn

utilization of deposited Zn during charging, thereby leading to a lower Coulombic efficiency (CE). Additionally, proton-induced corrosion, often characterized by hydrogen evolution, exacerbates the formation of 'dead' Zn, further diminishing the overall CE. Besides, the crossover of $Zn^{2+}$ from the negolyte to the posolyte also limits the cycling stability of ZIFBs. Consequently, the reported ZIFBs have seldom undergone cycles at current densities exceeding 80 mA cm$^{-2}$, as indicated in Supplementary Table 1. Achieving a prolonged cycle life has typically required the use of low current densities and/or additional Zn plating, which significantly compromises the inherent advantages of ZIFBs. Recent efforts have focused on optimizing the $Zn^{2+}$-negolyte to mitigate these aforementioned issues[18,19]. However, as of now, the attainment of a consistently stable cycle life at high current densities remains a rarity.

In this work, we introduce a $Zn(P_2O_7)_2^{6-}$ based negolyte, denoted as $Zn(PPi)_2^{6-}$ for simplicity, by directly chelating potassium pyrophosphate ($K_4P_2O_7$) with $ZnCl_2$. This negolyte is employed in the fabrication of ZIFBs. Through a combination of experimental data and theoretical calculations, we demonstrate that the $Zn(PPi)_2^{6-}$ based negolyte not only facilitates dendrite-free Zn plating but also effectively prevents $Zn^{2+}$ crossover. Consequently, ZIFB utilizing this approach exhibit remarkable stability even under high current density cycling. Remarkably, the use of chelated $Zn(PPi)_2^{6-}$ as a negolyte shifts the $Zn^{2+}$/Zn plating/stripping potential to −1.08 V (vs. SHE), resulting in an increased cell voltage of 1.61 V. This represents a 24% increase in cell voltage compared to conventional ZIFBs. Notably, this high-voltage

ZIFB demonstrates promising stability, exceeding 250 cycles at a high current density of 200 mA cm$^{-2}$ and a high power density up to 606.5 mW cm$^{-2}$.

## Results

Zinc-pyrophosphate chelated solution was obtained by adding $ZnCl_2$ solution dropwise to $K_4PPi$ solution with constant stirring (See Methods). To investigate the complexation ratio of the chelated solution, electrospray ionization-high resolution mass spectrometry (ESI-HRMS) was conducted. As depicted in Fig. 1a, an anion fragment peak of $[K_5ZnP_4O_{14}]^-$ (m/z = 606.5764) was clearly observed. Due to the presence of isotopes, the anion fragment peak of $[K_5ZnP_4O_{14}]^-$ were also observed at other positions such as m/z = 608.5736 and m/z = 610.5722. The fragment peaks of $[K_7ZnP_4O_{14}]^+$ (e.g., m/z = 684.5015) and $[K_4HZnP_4O_{14}]^-$ (e.g., m/z = 568.6188) have also been detected according to mass spectral data in Supplementary Fig. 1a and Fig. 1b. In addition, Cyclic voltammetry (CV) curves (Supplementary Fig. 2) of 0.05 M $Zn^{2+}$ mixed with different concentrations of $PPi^{4-}$ indicated that the slope of the equilibrium potential *vs.* ln[$P_2O_7^{4-}$] was close to 2, corresponding to a cation-anion coordination ratio of 1:2[20]. Based on the aforementioned findings, the chelated ion in the solution is proven to be $Zn(PPi)_2^{6-}$, and its coordination form is similar to other metal chelates comprising $P_2O_7^{4-}$ such as $Mn(P_2O_7)_2^{6-}$ and $Cu(P_2O_7)_2^{6-}$[21,22]. The structure of $Zn(PPi)_2^{6-}$ ion was further analyzed by $^{31}P$ nuclear magnetic resonance (NMR) spectra. As illustrated in Fig. 1b, a non-concentration-dependent single peak is observed at −5.8 ppm

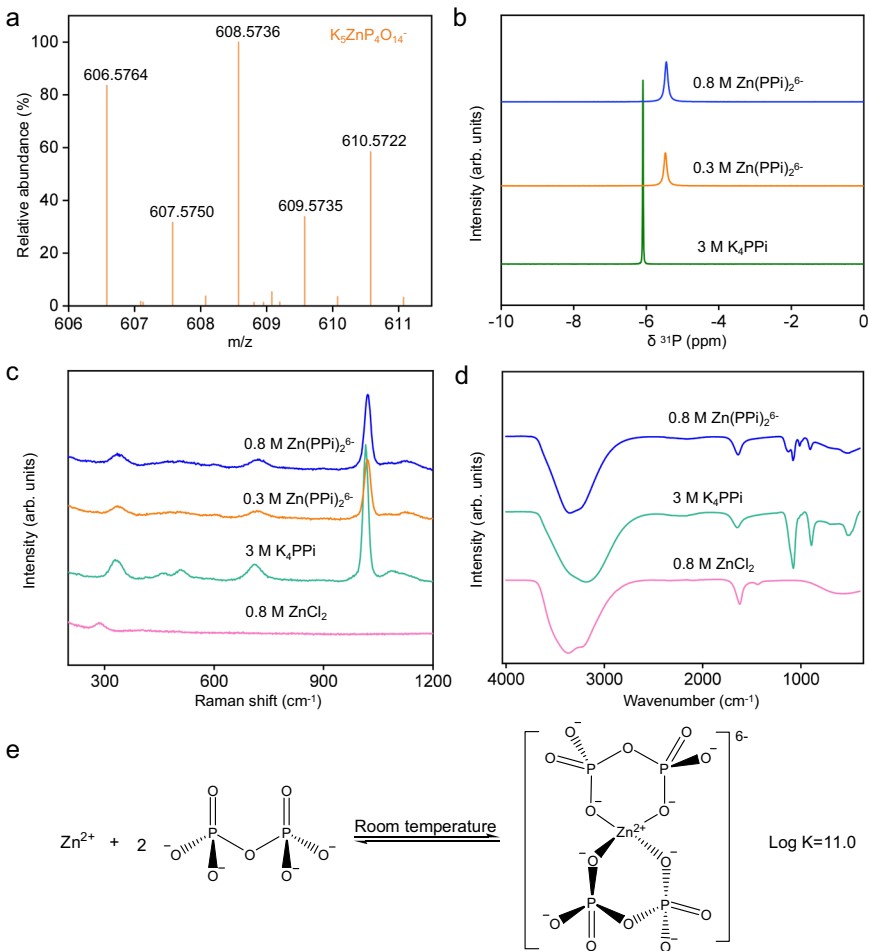

**Fig. 1 | Structural information of $Zn(PPi)_2^{6-}$ ions. a** ESI-HRMS spectrum of the $Zn(PPi)_2^{6-}$, the peak found at m/z = 606.5764 is assigned to $[K_5ZnP_4O_{14}]^-$ (calcd: 606.5715). **b** $^{31}P$ NMR of 3 M $K_4PPi$, 0.3 M and 0.8 M $Zn(PPi)_2^{6-}$, respectively. **c** Raman spectra of 0.8 M $ZnCl_2$, 3 M $K_4PPi$, 0.3 M and 0.8 M $Zn(PPi)_2^{6-}$, respectively. **d** ATR-FTIR spectra of 0.8 M $ZnCl_2$, 3 M $K_4PPi$ and 0.8 M $Zn(PPi)_2^{6-}$, respectively. **e** The chelated process of $Zn(PPi)_2^{6-}$ ions.

for the $Zn(PPi)_2^{6-}$ solution. For reference, the $^{31}P$ NMR spectrum of $K_4PPi$ solution with a symmetrical structure also shows a single peak at −6.2 ppm. This means that all P-atoms in $Zn(PPi)_2^{6-}$ are chemically equivalent. Besides, the chemical shift of $Zn(PPi)_2^{6-}$ solution exhibits a lower field shifting (toward higher ppm) compared with $K_4PPi$ solution, indicating the enhanced role of magnetic susceptibility[23], which is related to the decrease in charge density near the P atom after the introduction of $Zn^{2+}$ ion[24]. Meanwhile, Raman spectroscopy and attenuated total reflectance-fourier transform infrared (ATR-FTIR) spectroscopy were also conducted to uncover the structure of $Zn(PPi)_2^{6-}$ solution. As shown in Fig. 1c, the Raman peaks of $Zn(PPi)_2^{6-}$ solution detected at 1024 cm$^{-1}$ and 1140 cm$^{-1}$ belong to symmetric and antisymmetric stretching vibrations modes of $PO_3$ group[25,26], positively shifting compared with $K_4PPi$ parental solution (located at 1015 cm$^{-1}$ and 1090 cm$^{-1}$, respectively). Figure 1d presents the ATR-FTIR spectra of $ZnCl_2$, $K_4PPi$ and $Zn(PPi)_2^{6-}$, respectively. The vibrational peaks observed at 1012 cm$^{-1}$ and 1128 cm$^{-1}$ correspond to symmetric and antisymmetric stretching vibrations modes of $PO_3$ group in $Zn(PPi)_2^{6-}$, whereas the related vibrational peaks in $K_4PPi$ solution are located at 900 cm$^{-1}$ and 1078 cm$^{-1}$[26]. Combining the above experimental results and subsequent theoretical simulation in Fig. 2f, we depicted the formation process of $Zn(PPi)_2^{6-}$ in Fig. 1e. It is also found that the stability constant of $Zn(PPi)_2^{6-}$ is $1 \times 10^{11.027}$. Based on Le Chatelier-Braun's principle, excess $PPi^{4-}$ ligands promote the formation of the $Zn(PPi)_2^{6-}$ complex and improve its stability. When the concentration ratio of

$PPi^{4-}$ ligand to $Zn^{2+}$ is 3:1, the solubility of the $Zn(PPi)_2^{6-}$ solution reaches 0.9 M. As the molar ratio of the $PPi^{4-}$ to the $Zn^{2+}$ increases from 3:1 to 10:1, the conductivity of the resulting saturated solution gradually increases, whereas the solubility decreases from 0.9 M to 0.5 M (Supplementary Fig. 3). In addition, if $ZnBr_2$ is used to prepare the complex solution with a ligand-to-zinc ion ratio of 3:1, the solubility of the resulting complex solution will decrease to 0.7 M.

Molecular dynamics (MD) simulations were then carried out to analyze the solvation structure of the $ZnBr_2$ and $ZnCl_2$-$K_4PPi$ (1:3) systems, with the $Zn^{2+}$ concentration set at 0.2 M for both systems. Herein, we opted to utilize $ZnBr_2$ solution for comparative purposes, given its widespread application in zinc-based flow batteries. The results show that when the whole system became stable in a pure $ZnBr_2$ environment, six $H_2O$ molecules appeared in the primary solvation shell (PSS) of $Zn^{2+}$ (Fig. 2a), in accordance with previous literature[28]. On the contrary, when $PPi^{4-}$ was introduced into $ZnCl_2$, the PSS of $Zn^{2+}$ changed significantly (Fig. 2d). Analysis of the corresponding radial distribution functions (RDFs) and coordination numbers in different electrolytes show that for $ZnBr_2$ solution, a main peak of the Zn-O pair appeared at a distance of about 1.99 Å, which is attributed to $H_2O$ in the PSS (Fig. 2b), and the coordination number of $Zn^{2+}$ is near to 6. Meanwhile, for the $ZnCl_2$-$K_4PPi$ system, a sharp peak appeared at a distance of 1.7 Å from $Zn^{2+}$ with the O atoms in $PPi^{4-}$, and the coordination number of Zn-O in the first PSS is close to 4 (Fig. 2e), assuring the chelation of two $PPi^{4-}$ with one $Zn^{2+}$. Based on the results, density

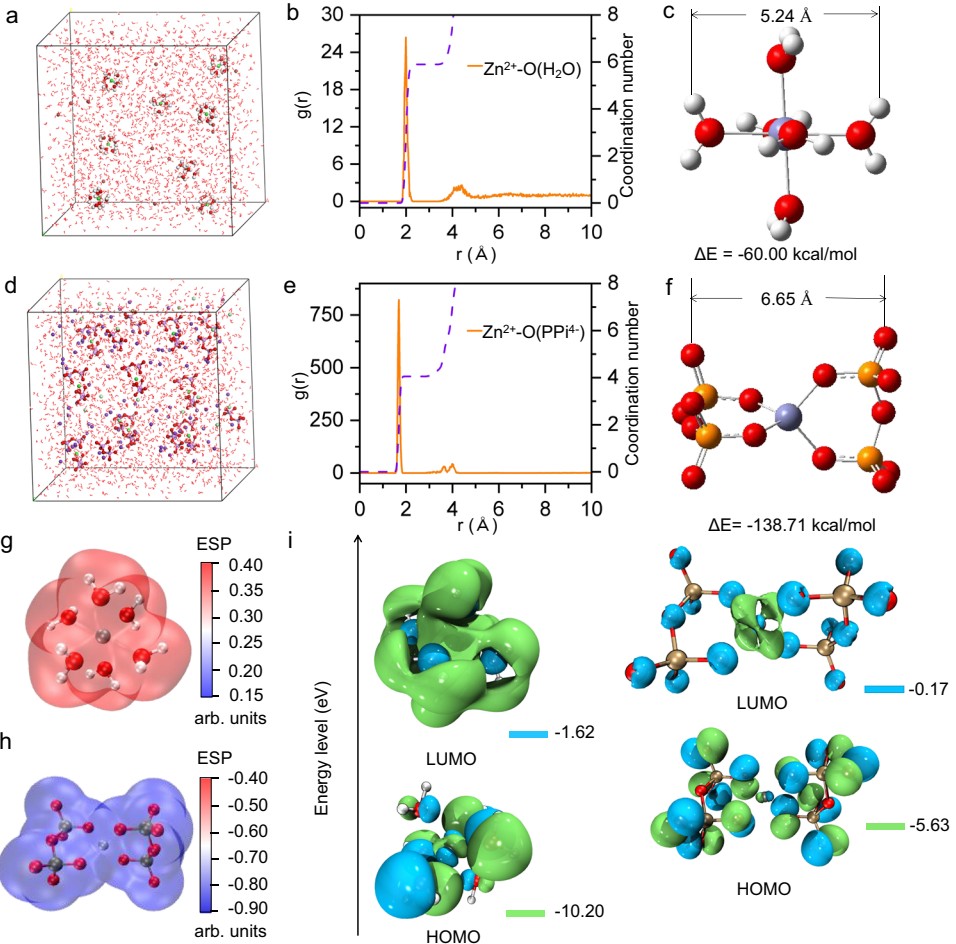

**Fig. 2 | Theoretical calculation results for $Zn(H_2O)_6^{2+}$ and $Zn(PPi)_2^{6-}$.** 3D snapshot of **a** 0.2 M $ZnBr_2$ system and **d** 0.2 M $ZnCl_2$-$K_4PPi$ (1:3) system obtained from MD simulations. RDFs for **b** $ZnBr_2$ and **e** $ZnCl_2$-$K_4PPi$ system collected from MD simulations. The optimized molecular structures and corresponding binding energy of **c** $Zn(H_2O)_6^{2+}$ and **f** $Zn(PPi)_2^{6-}$. ESP-mapped molecular van der Waals surface of **g** $Zn(H_2O)_6^{2+}$ and **h** $Zn(PPi)_2^{6-54}$. **i** The LUMO and HOMO isosurfaces of $Zn(H_2O)_6^{2+}$ (left) and $Zn(PPi)_2^{6-}$ (right), respectively.

functional theory (DFT) calculations were further conducted to gain insights into the interaction behavior between $Zn^{2+}$ ion and $H_2O$ or $PPi^{4-}$, respectively. The optimized structure of $Zn(H_2O)_6^{2+}$ is shown in Fig. 2c, and the binding energy of $Zn^{2+}$ with $H_2O$ is estimated to be -60.00 kcal mol$^{-1}$. The optimized structure of $Zn(PPi)_2^{6-}$ is shown in Fig. 2f, and the simulated infrared spectrum of $Zn(PPi)_2^{6-}$ is relatively close to the experimental spectrum (Supplementary Fig. 4). It is noteworthy that the bond length of P = O in $PPi^{4-}$ is 1.55 Å, whereas that of P = O in $Zn(PPi)_2^{6-}$ is 1.53 Å (Supplementary Fig. 5), which accounts for the blue-shifting in the infrared absorption after coordination (Fig. 1d)[29]. The binding energy of $Zn(PPi)_2^{6-}$ is estimated to be -138.7 kcal mol$^{-1}$, much higher than that of $Zn(H_2O)_6^{2+}$, indicating a stronger interaction of $Zn^{2+}$ with $PPi^{4-}$. Besides, $Zn(PPi)_2^{6-}$ shows a larger molecular size (6.65 Å) than that of $Zn(H_2O)_6^{2+}$ (5.34 Å). To understand the charge distribution and electron density of two species, electrostatic potential (ESP) mapped molecular van der Waals surfaces of them were also calculated. The results exhibit total different electric inherent, i.e., positive ESP for $Zn(H_2O)_6^{2+}$ ion (Fig. 2g), while negative ESP for $Zn(PPi)_2^{6-}$ ion (Fig. 2h). In general, the ion with positive ESP is electrophilic[30], and $Zn(H_2O)_6^{2+}$ has been shown to be susceptible to byproducts (e.g., $Zn(OH)_2$, ZnO, $Zn_5(OH)_8Cl_2 \cdot H_2O$, $Zn_4SO_4(OH)_6$, etc.) during Zn deposition process[31–33]. In contrast, $Zn(PPi)_2^{6-}$ with negative ESP is nucleophilic, which would contribute to the suppression of the by-product (e.g., $Zn(OH)_2$ and ZnO) formation during zinc deposition[32]. Frontier orbital analyses were conducted for two species to gain insight into the coordinator effect on the redox potential of $Zn^{2+}$, since the redox potential of an active molecule shows a substantial correlation with its LUMO energy level (Fig. 2i). The results show that $Zn(PPi)_2^{6-}$ owns a higher LUMO energy (-0.17 eV) than $Zn(H_2O)_6^{2+}$ (-1.62 eV), which is attributed to the stronger ligand basicity of $PPi^{4-}$ ions than water molecules[34].

To clarify the redox process of $Zn(PPi)_2^{6-}$, CV curves of 0.1 M $K_6Zn(PPi)_2$ electrolyte and 0.1 M $ZnBr_2$ electrolyte were measured on a carbon paper electrode (1 cm$^{-2}$) at 100 mV s$^{-1}$ for comparison, as shown in Fig. 3a. It is found that the plating/stripping potential of $Zn(PPi)_2^{6-}$/Zn is apparently negatively shifted to -1.08 V (vs. SHE), as supposed to

-0.76 V (vs. SHE) for $Zn^{2+}$/Zn. Note that the concentration of $Zn(H_2O)_6^{2+}$ is $1 \times 10^{-10}$ M in 0.1 M $Zn(PPi)_2^{6-}$ electrolyte based on the stability constant of $Zn(PPi)_2^{6-}$. Correspondingly, $\varphi_{Zn(PPi)_2^{6-}/Zn}$ is calculated to be -1.05 V (vs. SHE) using the Nernst equation, very close to our experimental value. This result indicates that the plating process of $Zn(PPi)_2^{6-}$ electrolyte consumes the free $Zn^{2+}$, and simultaneously the dissociation of $Zn(PPi)_2^{6-}$ releases the free $Zn^{2+}$. The kinetic rate constants of the two electrolytes were further investigated using the steady-state polarization method (Fig. 3b and Supplementary Fig. 6). The reduced rate constant ($k_0$) of $Zn^{2+}$ is determined to be $1.1 \times 10^{-4}$ cm s$^{-1}$. In contrast, the reduced rate constant ($k_0$) of $Zn(PPi)_2^{6-}$ is calculated to be $6.1 \times 10^{-5}$ cm s$^{-1}$, slightly lower than that of $Zn^{2+}$. In addition, the reduction peak of $Zn(PPi)_2^{6-}$ electrolyte is clearly visible compared to that of $Zn^{2+}$ electrolyte. To figure out the reason, CV curves at different sweep rates of $Zn(PPi)_2^{6-}$ electrolyte were investigated (Fig. 3c). The reduction peak current is linearly related to the square root of the sweep rates (Fig. 3d), and the diffusion coefficient ($D$) is calculated to be $3.03 \times 10^{-6}$ cm$^2$ s$^{-1}$ according to the Randles-Sevick equation[35], apparently lower than that of $Zn^{2+}$ ($2.44 \times 10^{-5}$ cm$^2$ s$^{-1}$)[36]. It is considered that the diffusion of $Zn(PPi)_2^{6-}$ to the electrode surface can't compensate for the $Zn^{2+}$ consumption, thus the cathodic current reaches its maximum value to form a reduction peak. This phenomenon also occurs in other $Zn^{2+}$-complex electrolytes, such as $ZnBr_4^{2-}$ and $Zn(NH_3)_4^{2+}$[19,37].

We firstly demonstrated two parallel near neutral ZIFBs, one with 0.2 M $ZnBr_2$ negolyte (pH=5.6), and the other with 0.2 M $K_6Zn(PPi)_2$ negolyte (pH=9.2), respectively. Both ZIFBs employed a low-cost polyolefin cation exchange membrane (JCM-D membrane) due to its lower area resistance (0.99 Ω cm$^2$) in 1 M KCl solution compared to Nafion 212 membrane (1.18 Ω cm$^2$), as shown in Supplementary Fig. 7. Figure 4a presents the galvanostatic charge/discharge (GCD) curves in the initial cycle for the two cells at a current density of 40 mA cm$^{-2}$. It is clearly observed that the ZIFB with $Zn(PPi)_2^{6-}$ negolyte not only exhibits higher cell voltage of 0.3 V than that of the ZIFB with $Zn^{2+}$ negolyte, but also shows a higher CE of 98% (79% for $Zn^{2+}$ negolyte). The fluctuation in the GCD curves of $Zn(PPi)_2^{6-}$ based ZIFB may be due to the disruption of the coordination equilibrium of $Zn(PPi)_2^{6-}$. Linear sweep

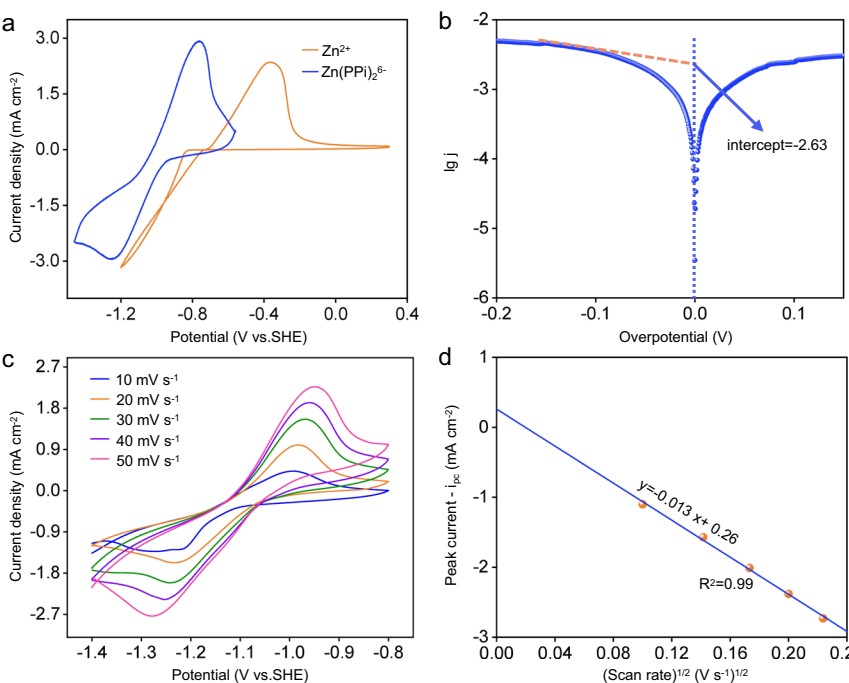

**Fig. 3 | Electrochemical properties of $Zn(PPi)_2^{6-}$ electrolyte. a** CV curves of 0.1 M $Zn(PPi)_2^{6-}$ and 0.1 M $ZnBr_2$ solution on a carbon paper electrode at 50 mV s$^{-1}$, respectively. **b** Tafel plots for Zn plating/stripping in 0.2 M $Zn(PPi)_2^{6-}$ solution at 0.1 mV s$^{-1}$. **c** CV curves of 0.1 M $Zn(PPi)_2^{6-}$ at various scan rates ranging from 10 to 50 mV s$^{-1}$. **d** Linear relationship between reduction peak current densities ($i_{pc}$) with square root of the scan rate ($v^{1/2}$) derived from **c**.

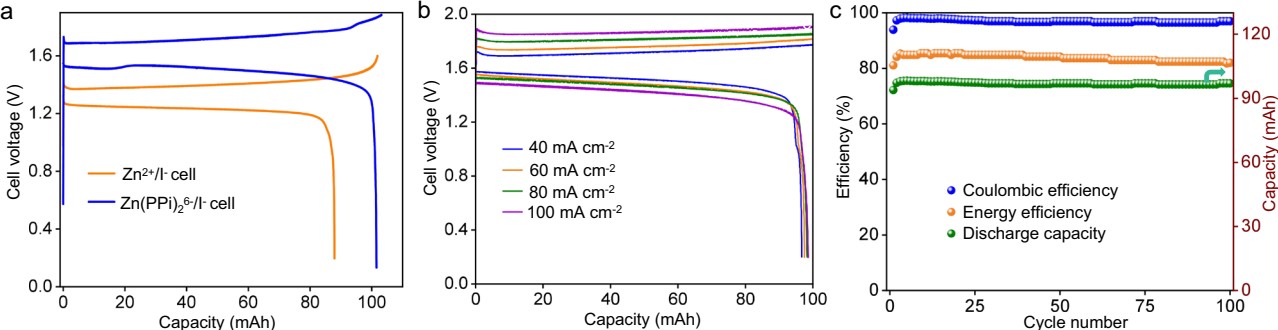

**Fig. 4 | Electrochemical performance of 0.2 M Zn(PPi)$_2^{6-}$ based ZIFBs. a** GCD profiles of the ZIFBs at 40 mA cm$^{-2}$ using 0.2 M Zn(PPi)$_2^{6-}$ negolyte or 0.2 M ZnBr$_2$ negolyte in the first cycle. The charge process ended with a cutoff voltage of 1.9 V and 1.6 V, respectively, while the discharge process ended with a cutoff voltage of 0.2 V. **b** Rate performance of 0.2 M Zn(PPi)$_2^{6-}$ based ZIFB with a charging capacity of 20 mAh cm$^{-2}$ at various current densities, the discharge process ended with a cutoff voltage of 0.2 V. **c** Cycling performance of 0.2 M Zn(PPi)$_2^{6-}$ based ZIFB at 40 mA cm$^{-2}$. The charging capacity was controlled to 20 mAh cm$^{-2}$, while the discharge process ended with a cutoff voltage of 0.2 V.

voltammetry (LSV) of the two negolytes was then performed, and the result showed that the hydrogen evolution reaction (HER) in the 0.2 M ZnBr$_2$ negolyte was more severe than that in the 0.2 M Zn(PPi)$_2^{6-}$ negolyte (Supplementary Fig. 8). The CV curves of cycled KI posolytes (Supplementary Fig. 9) indicates that Zn$^{2+}$ ions heavily penetrate the JCM-D membrane, whereas Zn(PPi)$_2^{6-}$ ions barely cross through the membrane (the reduction peak around -1.2 V is originated from KI electrolyte, as shown in Supplementary Fig. 10). Further crossover tests of Zn$^{2+}$ and Zn(PPi)$_2^{6-}$ ions through the JCM-D cation membrane were performed by H-type cells (Supplementary Fig. 11–14). The results show that the permeation rate of Zn$^{2+}$ ions is as high as $3.2 \times 10^{-3}$ cm$^2$ h$^{-1}$, but Zn(PPi)$_2^{6-}$ ions could not be detected in the reference cell, indicating that the chelated Zn(PPi)$_2^{6-}$ with multiple negative charges and larger molecular size could be isolated by the membrane. In addition, unlike Zn$^{2+}$ negolyte, which usually suffers from the hydrolysis side reaction in aqueous solution, Zn(PPi)$_2^{6-}$ negolyte exhibits an excellent chemical stability over three months, as revealed by the ATR-FITR spectra and GCD tests (Supplementary Fig. 15).

The electrochemical performance of Zn(PPi)$_2^{6-}$ based ZIFBs were further investigated. Supplementary Fig. 16a shows the relationship between open-circuit voltage (OCV) and state of charge (SOC) using 0.2 M Zn(PPi)$_2^{6-}$ negolyte, and the OCV gradually increases from 1.56 V to 1.68 V as the SOC increases from 10% to 100%. The best cell performance is achieved at 80% SOC in terms of energy efficiency and electrolyte utilization (Supplementary Fig. 17b). Subsequent rate and cycling performance tests were controlled with a deposited Zn areal capacity of 20 mAh cm$^{-2}$ (78% SOC). As shown in Fig. 4b, a slight increase in polarization potential difference is observed as the current density increases from 40 to 100 mA cm$^{-2}$, implying excellent energy efficiency of the cell. When the cell was cycled at 40 mA cm$^{-2}$, an average energy efficiency of 85% was achieved with negligible discharge capacity degradation over 100 cycles (Fig. 4c).

To investigate the plating/stripping behavior of the Zn(PPi)$_2^{6-}$ negolyte at high areal capacities, the assembled ZIFBs were charged and discharged at different depths using 0.8 M Zn(PPi)$_2^{6-}$ negolyte at a current density of 80 mA cm$^{-2}$. As the charge duration gradually increases from 0.5 to 2.5 h (Fig. 5a), the corresponding deposited Zn areal capacity increases from 40 to 180 mAh cm$^{-2}$. The discharge duration is always close to the charge duration, implying that the Zn(PPi)$_2^{6-}$ negolyte still maintains a high CE at high capacity deposition. In addition, the rate performance of the cell using 0.8 M Zn(PPi)$_2^{6-}$ negolyte was investigated (Fig. 5b), revealing a decrease in average energy efficiency from 87% to 65% as the current densities increased from 40 to 200 mA cm$^{-2}$. When the cell was cycled at high current density of 200 mA cm$^{-2}$ for 250 cycles, the representative GCD curves

at 10$^{th}$, 120$^{th}$ and 250$^{th}$ cycles showed a stable discharge voltage plateau near 1.4 V with no significant degradation in discharge capacity (Fig. 5c). The charging voltage polarization decreases gradually after a few cycles due to the close contact of the residual zinc with the carbon felt. The overall cycling performance of the cell is presented in Fig. 5d, which displays an average CE of over 97% and average energy efficiency around 70%. After the cycling tests, $^{31}$P NMR and ATR-FITR spectra of the Zn(PPi)$_2^{6-}$ negolyte were performed (Supplementary Fig. 17 and 18) and no new peaks were identified in either of the spectra, indicating its excellent electrochemical stability. The polarization curves of the cell at 30%, 50% and 80% SOCs conform a linear trend (Fig. 5e), which means that the voltage drop is dominated by the Ohmic polarization rather than the kinetic polarization. Benefiting from the high cell voltage of the ZIFB, it exhibits a maximum output power of 606.5 mW cm$^{-2}$ at 80% SOC, which is superior among reported ARFBs with cell voltages over 1.5 V[8,38-41]. When the deposited Zn areal capacity was increased to 60 mAh cm$^{-2}$, the cell still exhibited excellent performance with an average CE of 97% and an average energy density of 85% at 80 mA cm$^{-2}$ (Supplementary Fig. 19).

To visualize the comprehensive performance of the Zn(PPi)$_2^{6-}$ based ZIFBs, the deposited Zn areal capacity *vs.* current density is compared with other high-performance Zn-based flow batteries (ZRFBs)[10,15,18,19,31,42-45]. Figure 5f presents the areal capacity of several ZRFBs at different current densities. The observed high rate (i.e., 200 mA cm$^{-2}$) is superior to that of most reported ZRFBs. This phenomenon suggests that dissociation of Zn(PPi)$_2^{6-}$ (or the complexation of Zn$^{2+}$ and PPi$^{4-}$) occurs at a very rapid rate during the Zn-plating (or Zn-stripping) process. As previously mentioned, the stability constant (10$^{11}$) of Zn(PPi)$_2^{6-}$ indicates that the free Zn$^{2+}$ concentration is very low, at 10$^{-10}$ M, in the equilibrium state, signifying significant stability of the complex under equilibrium condition. However, it should be noted that such a high stability constant does not necessarily imply a low dissociation rate under non-equilibrium conditions. During the Zn-plating process, as free Zn$^{2+}$ is consumed, dissociation occurs rapidly to maintain the free Zn$^{2+}$ concentration at 10$^{-10}$ M. Similarly, during the Zn-stripping process, as the free Zn$^{2+}$ concentration increases, complexation occurs rapidly to ensure the concentration remains at 10$^{-10}$ M. Furthermore, the similar behavior has been observed in chelated ZnBr$_4^{2-}$ or Zn(OH)$_4^{2-}$ based zinc-iron flow batteries[19,46]. In addition, to demonstrate the ability of the Zn(PPi)$_2^{6-}$ electrolyte to be plated at high rate. We also performed a static three-electrode test (Supplementary Fig. 20), and it was found that the initial plating current density exceeded 140 mA cm$^{-2}$ when the polarization potential was controlled at -1.5 V vs. Ag/AgCl (i.e., -1.29 V vs. SHE), indicating a high dissociation kinetic of Zn(PPi)$_2^{6-}$. The rapid decrease in plating current density with

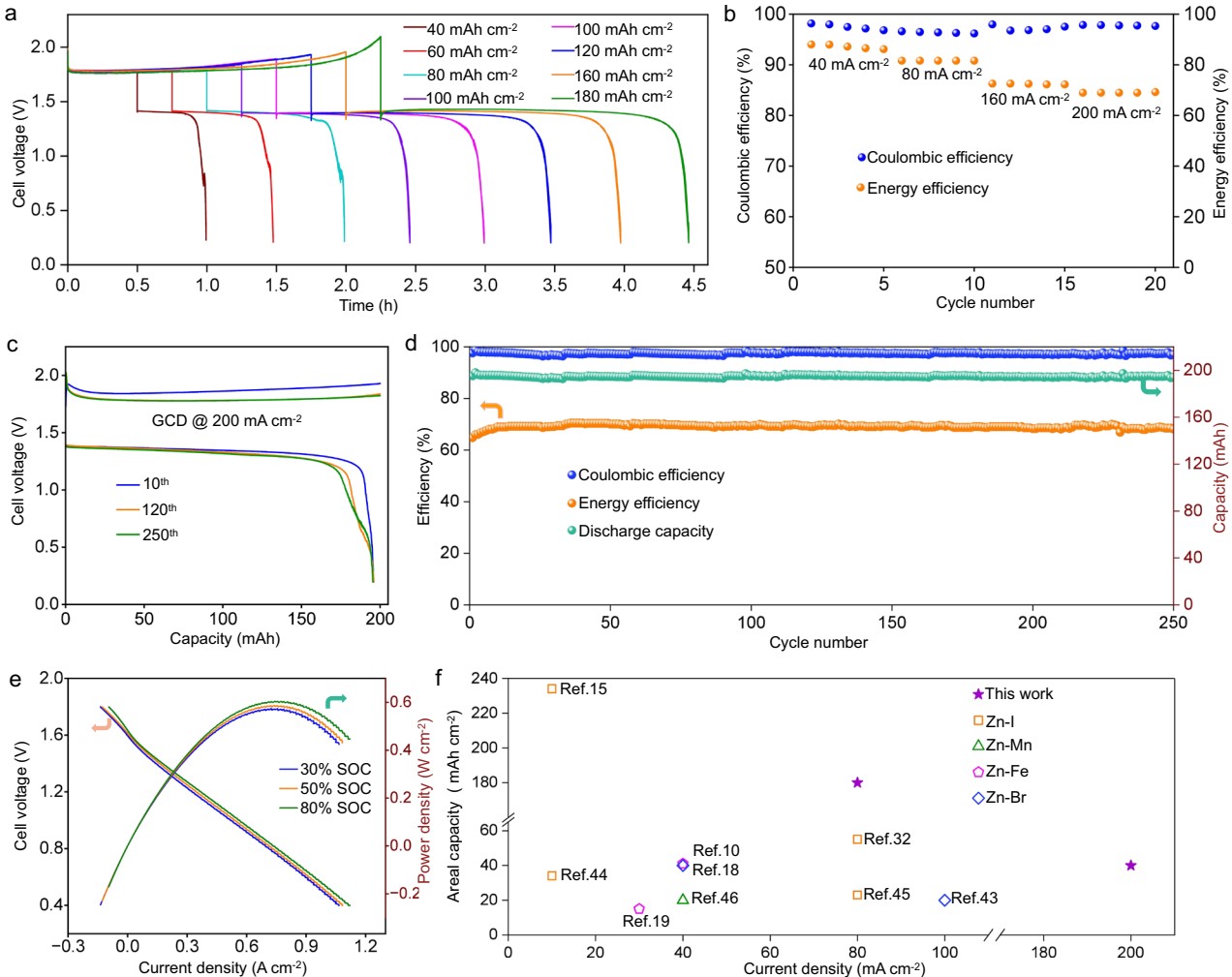

**Fig. 5 | Electrochemical performance of 0.8 M Zn(PPi)$_2^{6-}$ based ZIFBs. a** The GCD profiles of the ZIFB with different Zn areal capacities at 80 mA cm$^{-2}$. The charging capacity was controlled from 40 to 180 mAh cm$^{-2}$, while the discharge process ended with a cutoff voltage of 0.2 V. **b** Rate performance of the ZIFB at various current densities. The charging capacity was controlled to 200 mAh, while the discharge process ended with a cutoff voltage of 0.2 V. **c** Representative GCD curves of the ZIFB at 200 mA cm$^{-2}$. The charging capacity was controlled to 40 mAh cm$^{-2}$, while the discharge process ended with a cutoff voltage of 0.2 V. **d** Overall cycling performance of the ZIFB at 200 mA cm$^{-2}$. **e** Polarization and power curves of the ZIFB at different SOCs. **f** Performance comparison of several ZRFBs in terms of areal capacity and operating current density.

time suggests that the rate of diffusion of Zn(PPi)$_2^{6-}$ ions to the electrode surface determines the plating rate.

To gain insight into the mechanism of zinc deposition for Zn(PPi)$_2^{6-}$ negolyte, the Zn deposits on carbon felt using Zn$^{2+}$ negolyte or Zn(PPi)$_2^{6-}$ negolyte were collected by charging the corresponding ZIFBs to 100 mAh at 40 mA cm$^{-2}$ (Supplementary Fig. 21). Both of the zinc deposits were investigated by powder X-ray diffraction (PXRD), respectively (Supplementary Fig. 22, 23). The results show that the deposited Zn grows mainly along the (101) crystal plane for both negolytes. Nevertheless, the laser confocal scanning microscope (LCSM) image revealed an irregular Zn deposition morphology after Zn$^{2+}$ negolyte plating (Fig. 6a), and plenty of flower-like dendrites on the surface of carbon felt could be observed in the scanning electron microscopy (SEM) images (Supplementary Fig. 24a–c). However, for Zn(PPi)$_2^{6-}$ negolyte, the Zn deposits uniformly grew along carbon felt fibers (Fig. 6b), and no zinc dendrites were found in the SEM images (Supplementary Fig. 24 d-f). The element mapping with EDX analysis results also confirmed the even distribution of metallic Zn on carbon felt (Supplementary Fig. 25). To unveil the Zn growth mechanism of the two negolytes, we assembled symmetrical cells with uncharged filter paper as separator. The purpose of using uncharged filter paper is to exclude the Coulombic interaction between the membrane and the electrolyte, which could play a role in inhibiting Zn dendrite[38]. As shown in Supplementary Fig. 26, Zn(PPi)$_2^{6-}$ negolyte undergoes a higher nucleation overpotential (NOP) (220 mV) compared to Zn$^{2+}$ negolyte (30 mV), which is expected for a much smaller critical Zn nucleus radius (r), based on the relationship of r and NOP[47]:

$$r = 2\frac{\gamma V_m}{F|\eta|} \tag{1}$$

Where $\gamma$ is the surface energy of the Zn–electrolyte interface, $V_m$ is the molar volume of Zn, F is Faraday's constant, and $\eta$ is the NOP.

Subsequent SEM images of Zn deposits on carbon felt showed that the Zn$^{2+}$ negolyte tended to grow flower-like dendrites readily during the deposition process (Fig. 6c), whereas no dendrites were formed for Zn(PPi)$_2^{6-}$ negolyte plating (Fig. 6d). Therefore, it is reasonable to believe that the absence of dendrite upon the deposition of Zn(PPi)$_2^{6-}$ negolyte is primarily due to its high initial NOP on the carbon felt rather than the Coulombic repulsion between it and the anion

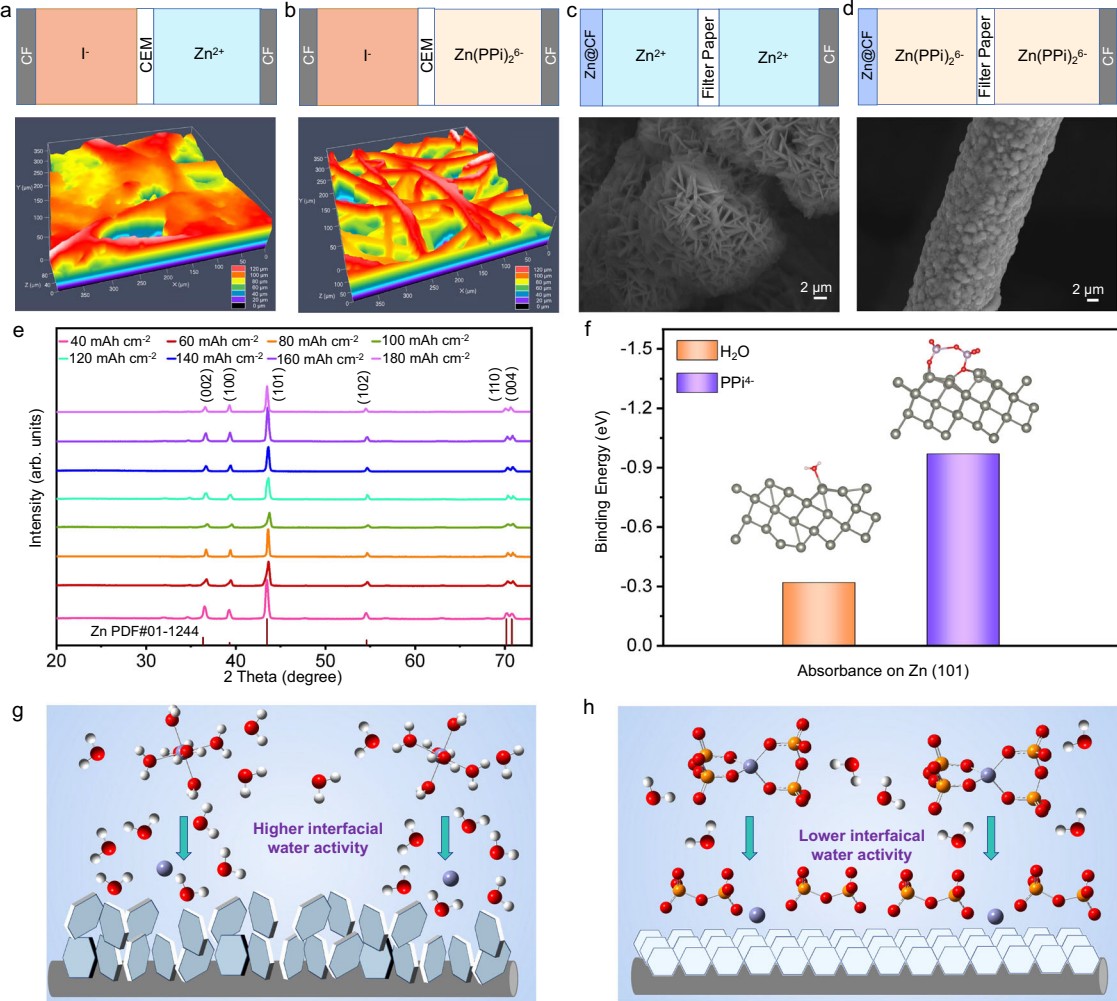

**Fig. 6 | Investigation of Zn deposition mechanism of two negolytes. a, b** Laser confocal scanning morphology of Zn deposits obtained by charging 0.2 M $Zn^{2+}$ negolyte and 0.2 M $Zn(PPi)_2^{6-}$ negolyte in an unsymmetrical ZIFB with a JCM-D CEM, respectively. **c, d** SEM morphology of Zn deposits obtained by charging 0.2 M $Zn^{2+}$ negolyte and $Zn(PPi)_2^{6-}$ negolyte in a symmetrical ZFB with a filter paper separator, respectively. **e** PXRD patterns of carbon felts for $Zn(PPi)_2^{6-}$ negolyte with deposition capacities ranging from 40 to 180 mAh $cm^{-2}$. **f** Binding energy of $H_2O$ molecule and $PPi^{4-}$ ion on the surface of Zn (101) crystalline plane. **g, h** The proposed Zn deposition process for $Zn^{2+}$ negolyte and $Zn(PPi)_2^{6-}$ negolyte, respectively.

groups on the JCM-D membrane. The PXRD patterns of carbon felts for $Zn(PPi)_2^{6-}$ negolyte with deposition capacities of 40-180 mAh $cm^{-2}$ (Fig. 6e) reveal the strongest Zn (101) crystalline diffraction peaks for all carbon felts, regardless of the deposition capacity. No zinc dendrites were observed on the surface of the carbon fibers despite the diameter of the fiber increased as the areal capacity increased (Supplementary Fig. 27). The binding energy for $H_2O$ and $PPi^{4-}$ on the Zn(101) crystalline plane was also carried out to analyze the dendrite-free growth mechanism for the $Zn(PPi)_2^{6-}$ electrolyte. It is found that the binding energy between a $PPi^{4-}$ ion and the uncharged Zn(101) surface is -1.05 eV (Fig. 6f), which is significantly higher than that between a $H_2O$ molecule and the uncharged Zn(101) surface (-0.32 eV). In addition, the zeta potentials of zinc powders in $H_2O$, $ZnBr_2$ and $K_4PPi$ solutions were evaluated. As depicted in Supplementary Fig. 28, the zeta potentials of Zn are -0.45 mV (in pure $H_2O$), 1.19 mV (in 0.8 M $ZnBr_2$), and -6.20 mV (in 0.8 M $K_4PPi$), respectively. The negative potential (-6.20 mV) strongly supports the calculation result that Zn metal prefers to adsorb $PPi^{4-}$. As mentioned above, the plating process of $Zn(PPi)_2^{6-}$ electrolyte consumes the free $Zn^{2+}$, and simultaneously the dissociation of $Zn(PPi)_2^{6-}$ releases the free $Zn^{2+}$. Unlike the dendrite deposition mode of conventional $Zn(H_2O)_6^{2+}$ ions on the Zn surface

(Fig. 6g) due to high interfacial water activity[48], the interfacial water activity of $Zn(PPi)_2^{6-}$ ions is effectively reduced, allowing the subsequent dissociated $Zn^{2+}$ ions to plate on the Zn surface in an orderly manner in assistance with the $PPi^{4-}$ ions (Fig. 6h). As a result, the $Zn(PPi)_2^{6-}$ electrolyte can alleviate the undesired HER and facilitate smooth Zn plating.

## Discussion

In summary, we have developed a chelated $Zn(PPi)_2^{6-}$ electrolyte with an impressive low plating/stripping potential (-1.08 V vs. SHE) via simply reacting cheap, non-toxic $K_4P_2O_7$ with highly soluble $ZnCl_2$. The strong interaction between ligands and zinc ions, the low redox potential and the fast reaction kinetics of $Zn(PPi)_2^{6-}$ were validated by corresponding spectra, electrochemical performances and DFT calculation. The designed $Zn(PPi)_2^{6-}$ negolyte coupled with KI posolyte could construct a dendrite-free ZIFB with a high cell voltage of 1.61 V. By employing a low-resistance polyolefin cation exchange membrane, the 0.8 M negolyte-based ZIFB can be operated stably at 200 mA $cm^{-2}$ over 250 cycles, and the excellent stability of the cycled negolyte is also confirmed by the $^{31}P$ NMR and ATR-FTIR spectra. By rationally designing the ligand structure to achieve an effective reduction in the

plating/stripping potential of the zinc complex ions, as well as preventing dendrite growth, this work provides a beneficial guide for the development of high-performance aqueous ZIFBs. Further endeavors should be exerted to achieve breakthroughs in energy density or long-term energy storage by combining high-potential electrolytes or enhancing cycling stability under conditions of ultra-high deposited zinc areal capacity.

## Methods

### Chemicals

Analytical grade Potassium iodide (KI, 99%) and Potassium pyrophosphate ($K_4PPi$, 99%) are purchased from Bide pharmatech Co., Ltd. Other reagents were purchased from Sinopharm Chemical Reagent Co. Ltd. All the reagents were used without further purification.

### Preparation of $Zn(PPi)_2^{6-}$ negolyte

$K_4PPi$ (39.64 g, 120 mmol) was dissolved in 25 mL of deionized water. Then, $ZnCl_2$ (5.452 g, 40 mmol) dissolved in 40 mL of deionized water was added dropwise to the $K_4PPi$ solution. The resulting chelated $Zn(PPi)_2^{6-}$ solution was stirred continuously until the solution became transparent, and then concentrated to 45 mL under reduced pressure at 50 °C. A low concentration of $Zn(PPi)_2^{6-}$ solution was obtained by diluting the saturated solution with deionized water.

### Materials characterizations

High-resolution mass spectra (HRMS) were obtained by Electrospray Ionization Mass Spectrometry (Bruker Dalton, maXis). $^{31}P$ NMR spectroscopy was carried out on Bruker DPX 400 MHz spectrometer, using $D_2O$ as the solvent. UV–vis spectra were measured using a UV–vis spectrometer (Shimadzu Scientific Instrument, UV-2600) and quartz spectrophotometer cells (Aldrich, 10 mm optical path length). Raman spectra were collected by a confocal Raman spectroscope (Horiba, LabRAM Evolution) with a 473 nm laser. Reflectance-Fourier transform infrared (ATR-FTIR) spectra were performed by a Cary 610/670 spectrometer instrument (Agilent Technologies Inc.). X-ray diffraction (XRD) patterns were acquired with a Bruker D8 ADVANCE X-ray diffractometer. The morphology of the as-deposited Zn on carbon felts were characterized by field-emission scanning electron microscopy (SEM, Carl Zeiss, Supra55) and confocal laser scanning microscope (CLSM, Carl Zeiss, LSM700). The ion conductivity of the negolytes were collected through a conductivity meter (Rex Electric Chemical, DDSJ-319L). Zeta potential measurements were performed using a Nanometrics instrument (ZEN3690).

### Theoretical computation methods

Molecular dynamics (MD) simulations were carried out using Material studio. Based on Newtonian mechanics, amorphous cell is used, where a certain percentage of particles are randomly placed inside the box (for 0.2 M $ZnBr_2$, $Zn^{2+}$:Br:$H_2O$ = 1:2:275 and for 0.2 M $ZnCl_2$-$K_4PPi$, $Zn^{2+}$:Cl:$K^+$:$PPi^{4-}$:$H_2O$ = 1:2:12:3:275). The number optimization iteration steps were set of 50,000 steps. Subsequently, an annealing operation was carried out using the Forcite module, with a low temperature setting of 300 K and a high temperature setting of 1000 K. The NPT system was used to eliminate the irrational conformation, and finally, a kinetic operation was carried out using Forcite, with a temperature setting of 298.15 K, running at 200 ps and the NPT system. Finally, the RDF data were collected.

DFT calculations for structural optimization of $Zn(H_2O)_6^{2+}$ and $Zn(PPi)_2^{6-}$ were carried out using the Gaussian 09 software package at the B3LYP levels with the 6−311 + G (d,p) basis set[49]. The polarizable continuum model (PCM) was employed in all calculations to account for the solvent effect in aqueous solution. For binding energy of Zn(101) surface with $H_2O$ molecule or $PPi^{4-}$, DFT calculations were carried out with the VASP code[50]. The Perdew−Burke−Ernzerhof (PBE) functional within generalized gradient approximation (GGA)[51] was used

to process the exchange–correlation, while the projectoraugmented-wave pseudopotential (PAW)[52] was applied with a kinetic energy cut-off of 500 eV, which was utilized to describe the expansion of the electronic eigenfunctions. The vacuum thickness was set to be 25 Å to minimize interlayer interactions. The Brillouin-zone integration was sampled by a Γ-centered 6 × 6 × 1 Monkhorst−Pack k-point. All atomic positions were fully relaxed until energy and force reached a tolerance of $1 \times 10^{-5}$ eV and 0.03 eV/Å, respectively. The dispersion corrected DFT-D method was employed to consider the long-range interactions[53].

### Electrochemical characterizations

Cyclic voltammetry (CV) curves and linear scanning voltammetry (LSV) curves were tested using an DH 7001 electrochemical workstation (Jiangsu Donghua Analytical Instrument Co., Ltd.) with a three-electrode system. A graphite rod (3 mm in diameter) and an Ag/AgCl electrode (pre-soaked in 3 M KCl solution) served as the counter electrode and reference electrode, respectively.

### Determination of the complex ratio

A commercial Zn foil (0.2 cm × 1 cm) was used as the working electrode (polished by the sandpaper). CV curves of the different chelated solutions were measured at a scan rate of 0.1 mV s$^{-1}$. The concentration of $Zn^{2+}$ is 0.05 mM, and the concentration ratio of $PPi^{4-}$ and $Zn^{2+}$ ranges from 10:1 to 30:1. In the case of a very slow sweep rate, it can be considered that the $Zn(P_2O_7)_m^{2-4m}$ solution and zinc electrode were kept in dynamic equilibrium, and the equilibrium equation could be described as follows:

$$Zn(P_2O_7)_m^{2-4m} + 2e^- \leftrightarrow Zn + mP_2O_7^{4-} \quad (2)$$

The complex-ratio $m$ can be calculated from the Nernst equation described as follows:

$$\varphi = \varphi_{Zn^{2+}/Zn} - \frac{RT}{2F} \ln \frac{\left[ P_2O_7^{4-} \right]^m}{\left[ Zn(P_2O_7)_m^{2-4m} \right]} \quad (3)$$

Where $\varphi$ is the equilibrium potential, $R$ is the gas constant: 8.314 J (mol K)$^{-1}$, $F$ is Faraday constant: 96485 C mol$^{-1}$. $T$ is the room temperature: 298.15 K.

Considering the constant concentration of $Zn(P_2O_7)_m^{2-4m}$, Eq. (3) can be transformed as follows:

$$\frac{\partial \varphi}{\partial \ln \left[ P_2O_7^{4-} \right]} = -\frac{mRT}{2F} \quad (4)$$

### Redox potential of the $Zn(PPi)_2^{6-}$ and $Zn^{2+}$

A piece of carbon paper (2 mm × 2.5 mm) was served as the working electrode. CV tests of 0.1 M $Zn(PPi)_2^{6-}$ or 0.1 M $ZnBr_2$ in 1 M KCl solution were performed with a sweep rate of 50 mV s$^{-1}$.

### The diffusion rate of $Zn(PPi)_2^{6-}$ ions

A piece of carbon paper (2 mm × 2.5 mm) was served as the working electrode. CV tests of 0.1 M $Zn(PPi)_2^{6-}$ in 1 M KCl solution were performed at different sweep rates from 10 to 50 mV s$^{-1}$. The diffusion coefficient is determined according to the Randles-Sevcik equation, as follows:

$$i_P = 0.4463 \, nFAC \left( \frac{nF\omega D}{RT} \right)^{1/2} \quad (5)$$

Where $i_P$ is peak current (mA), $n$ is number of electrons involved in the redox reaction, $F$ is Faraday constant: 96485 C mol$^{-1}$, $A$ is electrode area: 0.05 cm$^2$, $C$ is the concentration of active materials: 0.1 M, $R$ is the

gas constant: 8.314 J (mol K)$^{-1}$, $T$ is temperature: 298.15 K, $\omega$ is the scan rate (V s$^{-1}$), and $D$ is the diffusion coefficient (cm s$^{-1}$).

## Determination of the rate constant $k_0$ of $Zn^{2+}$ and $Zn(PPi)_2^{6-}$

A piece of carbon paper (2 mm × 1 mm) was served as the working electrode. LSV tests were conducted using 0.2 M $Zn^{2+}$ or 0.2 M $Zn(PPi)_2^{6-}$ in 1 M KCl solution with a sweep rate of 0.1 mV s$^{-1}$. The rate constant $k_0$ can be obtained from Eq. (6):

$$j = nFCk_0 \qquad (6)$$

where $j$ is the exchange current density (mA cm$^{-2}$), $n$ is the number of electrons involved in the redox reaction, $F$ is Faraday constant: 96485 C mol$^{-1}$, $C$ is the concentration of the electrolyte.

## The permeability of $Zn^{2+}$ and $Zn(PPi)_2^{6-}$ ions

The permeability of the active species through the JCM-D membrane (Cleanwater Technology Co., Ltd., Xiamen) was determined by a H-type cell. For the permeability measurement of $Zn^{2+}$, the left compartment of the diffusion cell was filled with 20 mL of 0.2 M $ZnBr_2$, while the right compartment was filled with 20 mL of 0.4 M KCl, and saturated zincon monosodium salt reagent was added to the right compartment in advance. For the permeability measurement of $Zn(PPi)_2^{6-}$, the left compartment of the diffusion cell was filled with 20 mL of 0.2 M $K_6Zn(PPi)_2$, while the right compartment was filled with 20 mL of 1.6 M KCl aqueous solution. The cell was continuously stirred during the measurements. The concentration of $Zn(PPi)_2^{6-}$ that permeated to the right compartment was monitored by a UV-vis spectrometer (Shimadzu Scientific Instrument, UV-2600). The permeability of $Zn^{2+}$ and $Zn(PPi)_2^{6-}$ electrolyte is calculated according to the Eq.(7):

$$P = \frac{\Delta\ln\left(1 - \frac{2c_t}{c_0}\right)\left(\frac{V_0 l}{2A}\right)}{\Delta t} \qquad (7)$$

where $P$ is the permeability of active species (cm$^2$ s$^{-1}$), $A$ is the effective area of the JCM-D membrane (0.2 cm$^2$), $c_t$ (mol L$^{-1}$) is the concentration of active species in the right cell at $t$, $c_0$ is the concentration of active species in the left cell (0.2 mol L$^{-1}$), $V_0$ is the volume of the solution in either reservoir (20 mL), $l$ is the thickness of the JCM-D membrane (30 $\mu$m thick).

## Battery performance

A single cell with an effective area of 2 cm × 2.5 cm was assembled by sandwiching a JCM-D membrane (30 $\mu$m thick) between two carbon felts (3 mm thick). The carbon felt were pre-heated in a furnace at 400 °C for 24 h and pressed onto the graphite flow fields on each side. The graphite plates were in contacted with two Cu current collectors, which were fixed by Acrylic end plates. The electrolytes were pumped into the corresponding flow fields by a peristaltic pump with two channels (Zibo Newking Electromechanical Equipment, China), with a flow rate of 60 mL min$^{-1}$. PTFE tubes (2 mm in diameter) were used to connect the cell. For paralled ZIFBs tests, 25 mL of 2 M KI + 1 M KCl and 10 mL of 0.2 M $Zn(PPi)_2^{6-}$ + 0.6 M KCl or 0.2 M $Zn^{2+}$ + 0.6 M KCl were used, respectively. For rate and cycling tests of 0.2 M $Zn(PPi)_2^{6-}$ based ZIFBs, 25 mL of 2 M KI + 1 M KCl and 12 mL of 0.2 M $Zn(PPi)_2^{6-}$ + 0.6 M KCl were used, respectively. For GCD profiles of the 0.8 M $Zn(PPi)_2^{6-}$ based ZIFB with different Zn areal capacities, 120 mL of 4 M KI + 2 M KCl and 25 mL of 0.8 M $Zn(PPi)_2^{6-}$ + 1.6 M KCl were used, respectively. For rate, cycling and polarization tests of 0.8 M $Zn(PPi)_2^{6-}$ based ZIFBs, 25 mL of 4 M KI + 2 M KCl and 6 mL of 0.8 M $Zn(PPi)_2^{6-}$ were used, respectively. The galvanostatic charge-discharge (GCD) tests were conducted at room temperature on a multichannel battery test system (Wuhan Land CT3002 AU). To obtain the cell power, polarization tests at different SOCs were performed (Jiangsu Donghua DH 7005) from 1.8 V to 0.4 V at a scan rate of 100 mV s$^{-1}$.

## Membrane area resistance tests

The cell was assembled with a membrane (Nafion 212 or JCM-D) and filled with 1 M KCl supporting salt. The effective area of the membrane is 2 cm × 2.5 cm. The resistance was measured using electrochemical impedance spectroscopy (EIS) over a frequency range from 10 HZ to 1 MHZ. The area resistance of the membrane can be calculated using the following equation:

$$R = (R_2 - R_1) \times S \qquad (8)$$

Where $R_1$ and $R_2$ are the resistance of the cell without and with a membrane, and $S$ is the effective area of a membrane.

## Data availability

The data that support the plots within this paper and other finding of this study are available from the corresponding author upon request.

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

## Acknowledgements

We acknowledge funding support from the National Natural Science Foundation of China (22209142, 22309160, 22225201), LiaoningBinhai-Laboratory (Grant No. LBLF-2023-05). Program of High-Level Innovation and Entrepreneurship Talents in Jiangsu Province, Lvyangjinfeng Talent Program of Yangzhou, and Natural Science Foundation of the Higher Education Institutions of Jiangsu Province (21KJB480006). We also thank Y.Q. Su at Xi'an Jiaotong University for assistance with DFT calculations and J. Xie for assistance with MD simulations. Supercomputing facilities were provided by Hefei Advanced Computing Center.

## Author contributions

C.X.W. conceived the idea for the project and synthesized the chelated electrolytes. Y.R.W. and Y.G.W. supervised the project. C.X.W., G.Y.G., and X.M.W. performed the electrochemical experiments. Y.Q.S performed the DFT calculations. J.X. performed the MD simulations. G.Y.G. and D.Y.H. performed electrolyte formulation and structural characterization. C.X.W., G.G.Y., Y.R.W. and Y.G.W. collectively wrote this manuscript. All the authors discussed the results and commented on the manuscript.

## Competing interests

The authors declare no competing interests.
