## [Peer Review File · Nature Communications]

High-Voltage and Dendrite-Free Zinc-Iodine Flow BatteryREVIEWER COMMENTS

Reviewer #1 (Remarks to the Author):

Manuscript 463657 entitled "High Voltage and Dendrite Free Zinc Iodine Flow Battery" by Wang and colleagues reports on the development of a Zinc-pyrophosphate chelate and its use as negative electrolyte material in Zinc-iodine flow batteries. The approach is interesting and the paper is fairly well written, but I am not convinced by the claimed mechanism. The authors state that the chelate enables dendrite free plating of zinc. I suspect, however, that the zinc complex remains in solution and acts a "regular" dissolved redox active molecule and that the zinc plating that is observed is simply from uncomplexed zinc in solution. I can therefore not recommend publication of the article. If the authors can provide compelling evidence about the zinc plating behavior the manuscript could be reconsidered.

Main concern:

L156: With such a strong binding energy, how is the chelate going to fall apart and plate zinc effectively? This is the first hint that you are not plating zinc but reducing the Zn-complex.

L175: Why would the plating/stripping potential decrease that much? At some point during the zinc plating reaction the complex would have to fall apart forming free zinc which could then be plated. This may be somehow assisted by the pyrophosphate but it would still be $Zn^{2+} + 2e^- \rightarrow Zn^0$.

Fig3a: this does not at all look like a plating stripping reaction. It looks much more like a "normal" CV of a redox active molecule in solution.

L258: Upon discharge any Zn^{2+} released from the anode would immediately be chelated by the pyrophosphate? It seems as if this process should take some amount of time, as described in your synthesis section.

Figure 5e: The intensity of plated Zinc is smaller at higher currents, this does not make any sense at all. Are you sure the "smooth" Zn you show in the SEM figure is not just uncomplexed Zn that is plated?

L322: Please explain why the 4- charged chelate is attracted more strongly to the negative electrode than a Zn^{2+} cation or water? Please explain how the water activity is decreased? If there are less Zn^{2+} ions that strongly bind water around themselves, how is the water activity lower?

To convince me, please quantify how much Zn you plate from the Zn-chelate solution.

Below are some additional remarks:

L40: the separation of energy and power components is largely lost with a plating/stripping electrode like zinc

L42: what structural features?

L44: VRFB have developed far beyond demonstration purposes with 100's of MWh installed globally

L52: corrosion of what?

L55: Why is Zn-dendrite formation more pronounced at high currents?

L57: Does the "dead Zn" dislodge and then flow through the cell? Does it clog the flowfields in any way?

L60: Can you explain the "H₂O induced corrosion"? Why is Zn^{2+} crossing into the positive side a problem, does Zn^{2+} react with the cathode?

Figure 1: I think it would help the reader to have some visual representation of the structure included in the MS/NMR/Raman/FTIR data.

L114: "meanwhile" not "in the meanwhile"

L118: Why do the PO₃ vibrations shift to higher wavenumbers? Can you explain what is going on here?

L125: Why did you chose a molar ratio of 3:1 and not 2:1?

L125: How did you determine the solubility?

L138: Why do you compare to ZnBr₂? Because of Zn-Br flow batteries? I think it would be good to spend 1-2 sentences explaining this.

L148: Is this concentration dependent?

L160: Please explain ESP with 1-2 sentences.

L165: The molecule has a 6- charge but it is electrophilic? How does that work? Why do you reference Fig 2d here?

L170: Can you elucidate why the LUMO of the complex is higher than the free Zn²⁺?

L204: Why is it described as if surprising that Zn²⁺ goes through a cation exchange membrane? What else would you expect?

Figure 4a: There are bumps in the blue curves at ca. 20 mAh and 90 mAh. What is going on there?

Figure 4f: Why is the charge polarization decreasing upon cycling?

L224: Again, why is it presented as surprising that the negatively charged chelate does not go through the cation exchange membrane?

L292: Coulombic interaction between the membrane and the electrolyte, why would this affect the dendrite formation?

L294: The high nucleation overpotential indicates that there should be a ca. 200 mV overpotential for plating Zn in the pyrophosphate solution? But you do not observe such a cell polarization?

L323: to analyze not to analysis

L335: Discussion  Conclusion: How does your work guide future efforts? Please explain better the key learnings and how they can guide future work.

Reviewer #2 (Remarks to the Author):

This manuscript presented a very nice work on negolyte development for zinc-iodine flow battery. Author's innovative refreshing chemistry by chelating K₄P₂O₇ with Zn²⁺ has produced a negolyte that enabled a high-voltage and dendrite-free zinc-iodine flow battery, performing significantly better than conventional zinc-iodine flow battery in terms of working current density and areal capacity. Their effort and novelty are to be commended, which will have important impact for flow battery technology. I recommend the manuscript to be accepted and published in Nature Communications. Some minor comments are as following:

1. For Zn(Pi)₂⁶⁻ based negolyte, why was ZnCl₂ employed, not ZnBr₂ (consistent with the results for Zn(H₂O)₆²⁺)?
2. For the permeability measurement of Zn²⁺, the right compartment was filled with 20 mL of 0.4 M KCl, while in Supplementary Fig. 11's caption, saturated zincon monosodium salt is added in the reference cell, is the zincon monosodium salt added into the right compartment prior to the permeability measurement?
3. How about the stability of Zn(Pi)₂⁶⁻ based negolyte at a high temperature?
4. In flow batteries tests, why excess posolyte is used?
5. Why the battery using Zn²⁺ negolyte shown a much low coulombic efficiency (79%)?
6. In line 194-202, Page 7, two parallel near neutral ZIFBs, 0.2 M ZnBr₂ negolyte (pH=5.6), and the other with 0.2 M K₆Zn(Pi)₂ negolyte (pH=9.2), were employed. Is this pH difference affecting CE of the battery? The HER potentials of ZnBr₂ negolyte and K₆Zn(Pi)₂ negolyte is suggested to provided.
7. In line 157, Page 6, Zn²⁺ has a stronger interaction with Pi⁴⁻, how about the desolvation process of Zn(Pi)₂⁶⁻ (or is the dissociation energy of Zn(Pi)₂⁶⁻ on anode surface high?)?

Response to Reviewer#1

Overall Comment: Manuscript 463657 entitled “High Voltage and Dendrite Free Zinc Iodine Flow Battery” by Wang and colleagues reports on the development of a Zinc-pyrophosphate chelate and its use as negative electrolyte material in Zinc-iodine flow batteries. The approach is interesting and the paper is fairly well written, but I am not convinced by the claimed mechanism. The authors state that the chelate enables dendrite free plating of zinc. I suspect, however, that the zinc complex remains in solution and acts a "regular" dissolved redox active molecule and that the zinc plating that is observed is simply from uncomplexed zinc in solution. I can therefore not recommend publication of the article. If the authors can provide compelling evidence about the zinc plating behavior the manuscript could be reconsidered.

Response: Thank you very much for reviewing our manuscript and giving many constructive comments. We would like to answer your questions separately and revise the manuscript according to your suggestions. All the revisions according to your questions/suggestions are marked in blue in the revised manuscript.

For your major concerns given in the overall comment, we give a response as follows:

(1) As correctly pointed out by you, the Zn plating process is based on the uncomplexed Zn^{2+} in the electrolyte, which is accompanied by the dissociation of $Zn(PPi)_2^{6-}$. In other words, the plating process consumes the free Zn^{2+} , and simultaneously the dissociation of $Zn(PPi)_2^{6-}$ releases the free Zn^{2+} .

Action: According to your comment, we have revised the manuscript as follows:

(Page 9 in the revised manuscript)

“This result indicates that the plating process of \$Zn(PPi)_2^{6-}\$ electrolyte consumes the free \$Zn^{2+}\$, and simultaneously the dissociation of \$Zn(PPi)_2^{6-}\$ releases the free \$Zn^{2+}\$.”

(2) According to the stability constant ($K = 10^{11.0}$ obtained from *Biosens. Bioelectron.* 50, 351-355) for $Zn(PPi)_2^{6-}$ [$Zn(PPi)_2^{6-} \leftrightarrow Zn^{2+} + 2PPi^{4-}$], we calculated the concentration of free Zn^{2+} in our electrolyte. For the 0.1 M $Zn(PPi)_2^{6-}$ negolyte (0.1 M $ZnCl_2$ + 0.3 M K_4PPi) used for CV test, we can assume the concentration of free Zn^{2+} as “ x ”, the concentration of $Zn(PPi)_2^{6-}$ as “0.1- x ”, and the concentration of PPi^{4-} as “0.1+2 x ” at the equilibrium state. Based on the equation of $\frac{0.1-x}{x(0.1+2x)^2} = 10^{11}$, we estimated the x -value to be 1×10^{-10} M. It means that the concentration of free Zn^{2+} in 0.1 M

Zn(PPi)₂⁶⁻ negolyte is 1×10⁻¹⁰ M. Considering the Nernst Equation, we believe such a low concentration (1×10⁻¹⁰ M) of free Zn²⁺ well explains the negative shift of Zn plating/stripping potential in the CV tests (see our response to question 2 for details).

Action: We have revised the manuscript as follows:

(Page 8~9 in the revised manuscript)

“Note that the concentration of Zn²⁺ is 1×10⁻¹⁰ M in 0.1 M Zn(PPi)₂⁶⁻ electrolyte based on the stability constant of Zn(PPi)₂⁶⁻. Correspondingly, $\phi_{\text{Zn(PPi)}_2^{6-}/\text{Zn}^{2+}}$ is calculated to be -1.05 V (vs. SHE) using the Nernst equation, which is very close to our experimental value.”

(3) The low concentration (~10⁻¹⁰ M) of free Zn²⁺ also alleviates undesired hydrogen evolution reaction (HER) during the Zn plating process, and consequently reduces the Zn dendrite growth. In conventional aqueous electrolytes containing large amounts of free Zn²⁺, the formation of hydrated Zn²⁺ is inevitable. During the Zn-plating process, some H₂O molecules in the solvated structure of Zn²⁺ obtain electrons, resulting in HER. Such undesired HER not only reduces the Coulombic efficiency, but also leads to inhomogeneous Zn plating. In the subsequent Zn plating, the top effect of these inhomogeneous Zn deposits will aggravate the dendrite growth. In the Zn(PPi)₂⁶⁻ electrolyte (e.g., 0.1 M ZnCl₂ + 0.3 M K₄PPi), the free Zn²⁺ is kept at a very low concentration (~ 10⁻¹⁰ M), and therefore H₂O molecules remain within the free solvent network or coordinate with K⁺ (1.2 M), rather than hydrated Zn²⁺ (Recently, Chunsheng Wang’s group has demonstrated a similar conclusion in their investigation about Zn plating/stripping behavior in a ZnCl₄²⁻ based electrolyte, *Nat. Sustain.* 2023, 6, 325-335). As a result, the Zn(PPi)₂⁶⁻ electrolyte can alleviate the undesired HER and facilitate smooth Zn plating.

Action: We have revised the manuscript as follows:

(Page 14 in the revised manuscript)

“As a result, the Zn(PPi)₂⁶⁻ electrolyte can alleviate the undesired HER and facilitate smooth Zn plating.”

Main concern:

Question-1: L156: With such a strong binding energy, how is the chelate going to fall apart and plate zinc effectively? This is the first hint that you are not plating zinc but reducing the Zn-complex.

Response: Thanks for your question. The response is given as follows:

(1) The Zn plating process depends on the free Zn^{2+} (i.e., the uncomplexed Zn^{2+}), which is accompanied by the dissociation of $Zn(PPi)_2^{6-}$. It means that the plating process consumes the free Zn^{2+} , and simultaneously the dissociation of $Zn(PPi)_2^{6-}$ releases the Zn^{2+} . In brief, the plating process does not involve the direct conversion between the chelated ($Zn(PPi)_2^{6-}$) and metallic Zn. The concentration of free Zn^{2+} in the electrolyte is controlled by the stability constant ($K = 1 \times 10^{11}$) for $Zn(PPi)_2^{6-}$. As shown in our response to your overall comment (or your question 2), the calculated concentration of free Zn^{2+} is $\sim 10^{-10}$ M in the $Zn(PPi)_2^{6-}$ electrolytes. Based on the Nernst equation, we calculated the Zn-plating potential at the concentration (free $[Zn^{2+}] \sim 1 \times 10^{-10}$ M), and obtained a redox potential of -1.05 V (vs. SHE), which is close to our experimental value of -1.08 V (vs. SHE) in the CV test (**Fig. 3a**). This result confirms that the Zn plating process depends on the free Zn^{2+} (i.e., uncomplexed Zn^{2+}).

(2) In our opinion, the fast dissociation of the Zn^{2+} -complex can provide enough free Zn^{2+} to ensure the Zn^{2+} consumption during the Zn plating process, which is confirmed not only by our experiments, but also by many previous reports. For example, in the widely reported and commercialized alkaline Ni-Zn batteries [i.e., $Ni(OH)_2/KOH$ electrolyte/Zn], the electrolyte contains a large amount of $Zn(OH)_4^{2-}$ and only trace amounts of free Zn^{2+} . However, in such batteries, the fast Zn-plating (i.e., the high rate charge of Ni-Zn batteries) has been well demonstrated (*J. Power Sources*, 2001, 100, 125-148). Some recent work has also demonstrated the Zn-plating process in Zn^{2+} -complex electrolytes, such as the $ZnCl_4^{2-}$ complex electrolyte (*Nat. Sustain.* 2023, 6, 325-335), and the EDTA- $Zn(OH)_3^-$ electrolyte (*Energy Environ. Sci.*, 2024, 17, 717-726).

Action: The related sentences are now provided in the revised manuscript as follows:

(Page 8~9 in the revised manuscript)

“Note that the concentration of Zn^{2+} is 1×10^{-10} M in 0.1 M $Zn(PPi)_2^{6-}$ electrolyte based on the stability constant of $Zn(PPi)_2^{6-}$. Correspondingly, $\phi_{Zn(PPi)_2^{6-}/Zn}$ is calculated to be -1.05 V (vs. SHE) using the Nernst equation, which is very close to our experimental value. This result indicates that the plating process of $Zn(PPi)_2^{6-}$ electrolyte consumes the free Zn^{2+} , and simultaneously the dissociation of $Zn(PPi)_2^{6-}$ releases the free Zn^{2+} .”

Question-2: L175: Why would the plating/stripping potential decrease that much? At some point

during the zinc plating reaction the complex would have to fall apart forming free zinc which could then be plated. This may be somehow assisted by the pyrophosphate but it would still be $\text{Zn}^{2+} + 2\text{e}^- \rightarrow \text{Zn}^0$.

Response: Thanks for your valuable question. We quite agree with you that the Zn plating process depends on the free Zn^{2+} ($\text{Zn}^{2+} + 2\text{e}^- \rightarrow \text{Zn}^0$), and the low concentration of the free Zn^{2+} in the Zn^{2+} -complex electrolyte solution should be the key reason for the negative shift plating potential. As mentioned in the response to your overall comment, the concentration of free Zn^{2+} in the electrolyte for the CV test is $\sim 10^{-10}$ M. According to the Nernst equation:

$$\Phi_{\text{Zn}(\text{PPI})_2^{6-}/\text{Zn}} = \Phi_{\text{Zn}^{2+}/\text{Zn}} + \frac{RT}{nF} \ln[\text{Zn}^{2+}] = -0.76 \text{ V} + 0.029 \log[\text{Zn}^{2+}]$$

the $\Phi_{\text{Zn}(\text{PPI})_2^{6-}/\text{Zn}}$ is calculated to be -1.05 V, which is very close to our experimental value (-1.08 V vs. SHE).

Action: We have revised the manuscript as follows:

(Page 8~9 in the revised manuscript)

“Note that the concentration of Zn^{2+} is 1×10^{-10} M in 0.1 M $\text{Zn}(\text{PPI})_2^{6-}$ electrolyte based on the stability constant of $\text{Zn}(\text{PPI})_2^{6-}$. Correspondingly, $\Phi_{\text{Zn}(\text{PPI})_2^{6-}/\text{Zn}}$ is calculated to be -1.05 V (vs. SHE) using the Nernst equation, which is very close to our experimental value. This result indicates that the plating process of $\text{Zn}(\text{PPI})_2^{6-}$ electrolyte consumes the free Zn^{2+} , and simultaneously the dissociation of $\text{Zn}(\text{PPI})_2^{6-}$ releases the free Zn^{2+} .”

Question-3: Fig 3a: this does not at all look like a plating stripping reaction. It looks much more like a "normal" CV of a redox active molecule in solution.

Response: Thank you very much for your good question. Yes, in the CV test with the conventional electrolyte containing a high concentration of free Zn^{2+} , the cathodic current for Zn-plating ($\text{Zn}^{2+} + 2\text{e}^- \rightarrow \text{Zn}^0$) continuously increases with negative sweep. This phenomenon is due to the fact that the free Zn^{2+} can diffuse from the electrolyte bulk to the electrode surface at a very high rate, which efficiently compensates for the Zn^{2+} consumption on the plating process. However, in the Zn^{2+} -complex electrolyte, the Zn plating process involves two steps: 1. The Zn^{2+} -complex diffusion from the bulk electrolyte to the near surface of the electrode, and 2. Dissociation of the Zn^{2+} -complex to release free Zn^{2+} for Zn plating. Generally, the diffusion rate of the Zn^{2+} -complex is lower than

that of the free Zn^{2+} . When the diffusion of the Zn^{2+} -complex cannot compensate the Zn^{2+} consumption, the cathodic current reaches its maximum value to form a reduction peak (i.e., the normal CV as you mentioned). For example, the similar CV curves have also been demonstrated in the $[ZnBr_4]^{2-}$ complex electrolyte (*Energy Storage Mater.*, 2022, 44, 433-440) and $Zn(NH_3)_4^{2+}$ complex electrolyte (*J. Electrochem. Soc.* 2017, 164 (4), D230-D236).

Action: We have revised the manuscript as follows:

(Page 9 in the revised manuscript)

“In addition, the reduction peak of $Zn(PPi)_2^{6-}$ electrolyte is clearly visible compared to that of Zn^{2+} electrolyte. To figure out the reason, the diffusion coefficient (D_0) of $Zn(PPi)_2^{6-}$ was investigated based on the CV curves at different sweep rates (**Fig. 3c**). The reduction peak current is linearly related to the square root of the sweep rates (**Fig. 3d**), and D_0 is calculated to be $3.03 \times 10^{-6} \text{ cm}^2 \text{ s}^{-1}$ according to the Randles-Sevcik equation,³⁶ which is apparently lower than that of Zn^{2+} ($2.44 \times 10^{-5} \text{ cm}^2 \text{ s}^{-1}$).³⁷ It is considered that the diffusion of $Zn(PPi)_2^{6-}$ to the electrode surface can't compensate for the Zn^{2+} consumption, thus the cathodic current reaches its maximum value to form a reduction peak. This phenomenon also occurs in other Zn^{2+} -complex electrolytes, such as $ZnBr_4^{2-}$ and $Zn(NH_3)_4^{2+}$.^{19, 38}”

The relative references have been also added in the revised manuscript as follows:

19. Yang, M.; Xu, Z.; Xiang, W.; Xu, H.; Ding, M.; Li, L.; Tang, A.; Gao, R.; Zhou, G.; Jia, C., High performance and long cycle life neutral zinc-iron flow batteries enabled by zinc-bromide complexation. *Energy Storage Mater.* **2022**, 44, 433-440.
38. Song, Y.; Hu, J.; Tang, J.; Gu, W.; Fu, Y.; Ji, X., The dynamic interfacial understanding of zinc electrodeposition in ammoniacal media through synchrotron radiation techniques. *J. Electrochem. Soc.* **2017**, 164 (4), D230-D236.

Question-4: L258: Upon discharge any Zn^{2+} released from the anode would immediately be chelated by the pyrophosphate? It seems as if this process should take some amount of time, as described in your synthesis section.

Response: Thank you for your question. Yes, the released Zn^{2+} from the anode is indeed immediately chelated by the high concentration of PPi^{4-} due to the high stability constant of $Zn(PPi)_2^{6-}$, which makes the free Zn^{2+} at a very low concentration of $\sim 10^{-10} \text{ M}$. As you mentioned,

we have ever described the electrolyte preparation as “ZnCl₂was added dropwise to the K₄PPi solution”. In fact, we slowly added 1 M ZnCl₂ into the excess PPI⁴⁻ to prevent Zn₂PPI precipitation which is caused by localized over-concentration of Zn²⁺. In the electrochemical process (i.e., the discharge process), Zn²⁺ is released continuously in the high flowing electrolyte without the issue of localized over-concentration.

Question-5: Figure 5e: The intensity of plated Zinc is smaller at higher currents, this does not make any sense at all. Are you sure the "smooth" Zn you show in the SEM figure is not just uncomplexed Zn that is plated?

Response: Thank you for your good question. Herein we response your question as follows:

(1) We believe that all the smooth Zn is derived from the free Zn²⁺ (i.e., uncomplexed Zn) provided by the dissociation of Zn(PPI)₂⁶⁻. In other words, the plating consumes the free Zn²⁺, and simultaneously the dissociation of Zn(PPI)₂⁶⁻ releases the free Zn²⁺. In the response to your overall comment, we have explained that the stability constant (K) for Zn(PPI)₂⁶⁻ results in the presence of free Zn²⁺ at a low concentration, which facilitate smooth Zn plating (Please see our response to your overall comment for detailed discussion).

(2) Fig. 5e illustrates the PXRD of deposited Zn at a constant current of 80 mA cm⁻², encompassing areal capacities ranging from 40 to 180 mAh cm⁻². Please note that we employ different carbon felts for separate Zn deposition, and therefore the relative intensities of the peaks may vary slightly. This variation could be associated with the differing degrees of compression in the carbon felts used each time.

Question-6: L322: Please explain why the 4- charged chelate is attracted more strongly to the negative electrode than a Zn²⁺ cation or water? Please explain how the water activity is decreased? If there are less Zn²⁺ ions that strongly bind water around themselves, how is the water activity lower ?

Response: Thank you for your good questions. Herein, we explain the ion adsorption on the pristine Zn surface (at the open circuit potential, OCP, without any charge), the ion adsorption on the Zn surface during the plating process (at the plating potential with electrons), and the water activity, respectively. See the follow points (1~3) for details.

(1) Yes, in our previous submission (L322 ~ L326), we have mentioned that the pristine Zn

(without any charge) prefers to adsorb PPi^{4-} , which is supported by the calculation (**Figure 5f**). As shown in **Figure 5f**, the calculated adsorption energies of PPi^{4-} and H_2O molecules on the surface of Zn(101) are -1.05 eV and -0.32 eV, respectively. It should be noted that the calculation is based on pristine Zn without any charge. In response to your question, we have compared the zeta potentials of zinc powder suspended in pure H_2O , 0.8 M ZnBr_2 , and 0.8 M K_4PPi in the revised supporting information (see the New **Supplementary Fig. 27**). As depicted in **Supplementary Fig. 27**, the zeta potentials of Zn are -0.45 mV (in pure H_2O), 1.19 mV (in 0.8 M ZnBr_2), and -6.20 mV (in 0.8 M K_4PPi), respectively. The negative potential (-6.20 mV) strongly supports the calculation that pristine Zn prefers to adsorb PPi^{4-} . Such adsorption should be helpful for the formation of uniform nucleation on initial plating process. If the Zn surface adsorbs a lot of H_2O molecules or hydrated Zn^{2+} , the initial Zn-plating might involve electrochemical reduction of H_2O (to generate H_2), resulting inhomogeneous nucleation.

Action: The related sentences have been added in revised the manuscript and supporting information as follows:

(Page 14 in the revised manuscript)

“In addition, the zeta potentials of zinc powders in H_2O , ZnBr_2 and K_4PPi solutions were evaluated. As depicted in **Supplementary Fig. 27**, the zeta potentials of Zn are -0.45 mV (in pure H_2O), 1.19 mV (in 0.8 M ZnBr_2), and -6.20 mV (in 0.8 M K_4PPi), respectively. The negative potential (-6.20 mV) strongly supports the calculation result that pristine Zn prefers to adsorb PPi^{4-} .”

(Page 16 in the revised manuscript)

“Zeta potential measurements were performed by Nanometrics (ZEN3690).”

(Page 28 in the revised supporting information)

Supplementary Fig. 27 Zeta potential of zinc powder in various solutions.

(2) During the plating process, the Zn electrode gains electrons (i.e., becomes negatively charged), and the ion adsorption is illustrated in **Figure Answer 1**. As depicted in **Figure Answer 1**, the surface adsorption on the Zn electrode (with negative charges) includes the inner layer of free Zn^{2+} and the outer layer of ZnPPi_2^{6-} . The plating process consumes the free Zn^{2+} in the inner layer, while simultaneously, the outer layer of ZnPPi_2^{6-} provides the necessary free Zn^{2+} through a dissociation reaction.

Figure Answer 1. Schematic illustration of the ions adsorption on the anode during the plating process.

(3) In conventional aqueous electrolytes containing Zn^{2+} , the Zn^{2+} ions are surrounded by H_2O molecules, forming hydrated Zn^{2+} complexes, such as $[\text{Zn}(\text{H}_2\text{O})_6]^{2+}$. During the Zn-plating process, some H_2O molecules in the solvation structure of Zn^{2+} gain electrons, leading to the occurrence of the hydrogen evolution reaction (HER). This indicates that the H_2O in the solvation structure of Zn^{2+} is highly active at the interface. In contrast, in the $\text{Zn}(\text{PPi})_2^{6-}$ electrolyte, the concentration of free Zn^{2+} remains very low (approximately $\sim 10^{-10}$ M). Consequently, H_2O molecules remain within the free solvent network or coordinate with K^+ , rather than forming hydrated Zn^{2+} . Notably, Chunsheng Wang's group recently demonstrated a similar conclusion in their investigation of Zn plating/stripping behavior in a ZnCl_4^{2-} based electrolyte (*Nat. Sustain.* 6, 325–335, 2023).

Action: We have revised the manuscript as follows:

(Page 14 in the revised manuscript)

“As mentioned above, the plating process of $\text{Zn}(\text{PPi})_2^{6-}$ electrolyte consumes the free Zn^{2+} , and simultaneously the dissociation of $\text{Zn}(\text{PPi})_2^{6-}$ releases the free Zn^{2+} . Unlike the dendrite deposition mode of conventional $\text{Zn}(\text{H}_2\text{O})_6^{2+}$ ions on the Zn surface (**Fig. 5g**) due to high interfacial water

activity,⁴⁸ the interfacial water activity of $\text{Zn}(\text{PPi})_2^{6-}$ ions is effectively reduced, allowing the subsequent dissociated Zn^{2+} ions to plate on the Zn surface in an orderly manner in assistance with the PPi^{4-} ions (**Fig. 5h**).”

Question-7: To convince me, please quantify how much Zn you plate from the Zn-chelate solution.

Response: Due to the high stability constant of $\text{Zn}(\text{PPi})_2^{6-}$, the mole fraction of $x_{\text{Zn}(\text{PPi})_2^{6-}} = \frac{[\text{Zn}(\text{PPi})_2^{6-}]}{[\text{Zn}^{2+}] + [\text{Zn}(\text{PPi})_2^{6-}]} \approx 1$. It means that all the Zn deposition arises from the $\text{Zn}(\text{PPi})_2^{6-}$ anions. However, as we mentioned above, the Zn plating process is based on the uncomplexed Zn^{2+} in the electrolyte, which is accompanied by the dissociation of $\text{Zn}(\text{PPi})_2^{6-}$. In other words, the plating consumes the free Zn^{2+} , and simultaneously the dissociation of $\text{Zn}(\text{PPi})_2^{6-}$ releases the free Zn^{2+} .

Below are some additional remarks:

Question-8: L40: the separation of energy and power components is largely lost with a plating/stripping electrode like zinc

Response: We acknowledge that the adjusting ability of Zinc-based flow batteries (ZFBs) for energy and power separation is lower compared to fully soluble flow batteries, such as All-vanadium flow batteries. However, the operational flexibility of ZFBs still surpasses that of conventional rechargeable batteries in adjusting power and energy. Notably, the environmentally friendly and low-cost Zinc-based electrolyte makes ZFBs an attractive option. As a result, ZFBs are garnering extensive attention and have successfully demonstrated applications at the MW/MWh level (*Mater. Today Energy*, 2018, 8, 80-108).

Action: According to you question, we have given a sentence to explain the advantage of ZFBs in the revised manuscript as follows:

(Page 3 in the revised manuscript)

“In addition to the fully soluble ARFBs mentioned above, zinc-based flow batteries have also made great strides in scaled energy storage due to the inexpensive zinc electrolyte, which can now reach the MW/MWh level.¹²”

The relative reference has been also added in the revised manuscript as follows:

12. Khor, A.; Leung, P.; Mohamed, M. R.; Flox, C.; Xu, Q.; An, L.; Wills, R. G. A.; Morante, J. R.; Shah, A. A., Review of zinc-based hybrid flow batteries: From fundamentals to applications. *Mater. Today Energy* **2018**, *8*, 80-108.

Question-9: L42: what structural features?

Response: Thanks for your question. ARFBs features excellent scalability, modular manufacturing and flexible design.

Action: We have revised the related sentence in the revised manuscript as follows:

(Page 3 in the revised manuscript)

“Moreover, ARFBs can decouple power and energy, all while meeting stringent safety requirements due to the features of excellent scalability, modular manufacturing, flexible design, as well as the non-flammability of aqueous electrolytes.⁵”

Question-10: L44: VRFB have developed far beyond demonstration purposes with 100's of MWh installed globally

Response: Yes, VRFBs are in the commercialization stage.

Action: We have revised the corresponding discussion in the revised manuscript as follows:

(Page 3 in the revised manuscript)

“As an illustration, all-vanadium ARFBs are currently the most widely commercialized RFB system.⁶”

The relative reference has been also added in the revised manuscript as follows:

6. Sun, C.; Zhang, H., Review of the development of first-generation redox flow batteries: Iron-chromium system. *ChemSusChem*, **2022**, *15* (1), e202101798.

Question-11: L52: corrosion of what?

Response: Thank you for your question. In mild aqueous electrolytes with a pH value close to 7 (ranging from 3 to 11, for example), the presence of protons (H^+) may induce Zn corrosion through the chemical reaction: $Zn + 2H^+ \rightarrow Zn^{2+} + H_2$. It appears that the term 'H₂O-induced corrosion' may not accurately describe the process and has been replaced with 'proton-induced corrosion'.

Action: We have revised the corresponding description in the revised manuscript as follows:

(Page 3 in the revised manuscript)

“Additionally, proton-induced corrosion, often characterized by hydrogen evolution, exacerbates the formation of 'dead' Zn, further diminishing the overall CE. Besides the challenges posed by Zn-dendrite growth and proton-induced corrosion, the crossover of Zn^{2+} from the negolyte to the posolyte also limits the cycling stability of ZIFBs.”

Question-12: L55: Why is Zn-dendrite formation more pronounced at high currents?

Response: Thanks for your good question. Zn^{2+} is deposited in the tip region as a result of an uneven electric field on the surface of the zinc anode. The higher the current density, the faster Zn^{2+} is deposited, which leads to more severe concentration polarization and thus faster growth of zinc dendrites (*ChemSusChem* 2018, 11, 3996-4006).

Action: We have mentioned this point in the revised manuscript as follows:

(Page 3 in the revised manuscript)

“It is widely recognized that the growth of Zn-dendrites on the anode becomes more severe at higher charging current densities (in $mA\ cm^{-2}$), thereby elevating the risk of short circuits.¹⁷”

The relative reference has been also added in the revised manuscript as follows:

17. Lu, W.; Xie, C.; Zhang, H.; Li, X., Inhibition of zinc dendrite growth in zinc-based batteries. *ChemSusChem* 2018, 11 (23), 3996-4006.

Question-13: L57: Does the "dead Zn" dislodge and then flow through the cell? Does it clog the flowfields in any way?

Response: Thank you for your good questions. Yes, "dead Zn" can detach from the carbon felt and even clog the tube, leading to a rapid loss of capacity. We performed the cycling test of 0.2 M $ZnBr_2$ based ZIFB, which couldn't work after several cycles. When we disassembled the cell, "dead Zn" clog the outlet hole, and plenty of loose "dead Zn" was found on the JCM-D membrane, as shown in

Figure Answer 2.

Figure Answer 2. a, Photograph of "dead Zn" blocking the outlet hole. b, Photograph of "dead Zn" on the JCM-D membrane.

Question-14: L60: Can you explain the "H₂O induced corrosion"? Why is Zn²⁺ crossing into the positive side a problem, does Zn²⁺ react with the cathode?

Response: Herein we answer your questions as follows:

(1) In mild aqueous electrolytes with a pH value close to 7 (ranging from 3 to 11, for example), the presence of protons (H⁺) may induce Zn corrosion through the chemical reaction: $\text{Zn} + 2\text{H}^+ \rightarrow \text{Zn}^{2+} + \text{H}_2 \uparrow$. It appears that the term 'H₂O-induced corrosion' may not accurately describe the process and has been replaced with 'proton-induced corrosion'.

(2) Zn²⁺ ions shuttling to the catholyte do not react with the catholyte, but since Zn²⁺ ions can't migrate back quickly enough to the negolyte, it will result in a decrease of the total Zn²⁺ in the negolyte when zinc is deposited on the anode.

Action: We have revised the corresponding description in the revised manuscript as follows:

(Page 3 in the revised manuscript)

“Additionally, proton-induced corrosion, often characterized by hydrogen evolution, exacerbates the formation of 'dead' Zn, further diminishing the overall CE. Besides the challenges posed by Zn-dendrite growth and proton-induced corrosion, the crossover of Zn²⁺ from the negolyte to the posolyte also limits the cycling stability of ZIFBs.”

Question-15: Figure 1: I think it would help the reader to have some visual representation of the structure included in the MS/NMR/Raman/FTIR data.

Response: Thank you for four good suggestion.

Action: We have added the appropriate reaction route in the revised **Fig. 1e** in the revised manuscript

as follows:

(Page 5 in the revised manuscript)

Fig. 1 Structure information of $\text{Zn}(\text{PPi})_2^{6-}$ ions. **a** HRMS spectrum of the $\text{Zn}(\text{PPi})_2^{6-}$, the peak found at $m/z=606.5764$ is assigned to $[\text{K}_5\text{ZnP}_4\text{O}_{14}]^-$ (calcd: 606.5715). **b** ^{31}P NMR spectra of 3 M K_4PPi , 0.3 M and 0.8 M $\text{Zn}(\text{PPi})_2^{6-}$, respectively. **c** Raman spectra of 0.8 M ZnCl_2 , 3 M K_4PPi , 0.3 M and 0.8 M $\text{Zn}(\text{PPi})_2^{6-}$, respectively. **d** FTIR spectra of 0.8 M ZnCl_2 , 3 M K_4PPi and 0.8 M $\text{Zn}(\text{PPi})_2^{6-}$, respectively. **e** The chelated process of $\text{Zn}(\text{PPi})_2^{6-}$ ions.

(Page 6 in the revised manuscript)

“Combining the above experimental results and subsequent theoretical simulations in Fig. 2, we depicted the formation process of $\text{Zn}(\text{PPi})_2^{6-}$ in Fig. 1e. It is also found that the stability constant of $\text{Zn}(\text{PPi})_2^{6-}$ is $1 \times 10^{11.0}$.²⁷”

The relative reference has been also added in the revised manuscript as follows:

27. Kong, R.-M.; Fu, T.; Sun, N.-N.; Qu, F.-L.; Zhang, S.-F.; Zhang, X.-B., Pyrophosphate-regulated Zn^{2+} -dependent dnzyme activity: An amplified fluorescence sensing strategy for alkaline phosphatase. *Biosens. Bioelectron.* **2013**, *50*, 351-355.

Question-16: L114: "meanwhile" not "in the meanwhile"

Response: Thank you for your suggestion. We have corrected this phrase in the revised manuscript as follows:

Action: We have corrected this phrase in the revised manuscript.

Question-17: L118: Why do the PO_3 vibrations shift to higher wavenumbers? Can you explain what is going on here?

Response: Thank you for your good question. The blue-shifting effects of the central metal ion on the ligand have been reported due to the decrease in the bond length of the ligand. See *J. Am. Chem. Soc.* 1956, 78, 14, 3295-3297.

For $\text{Zn}(\text{PPi})_2^{6-}$, Zn^{2+} has a pulling electron effect on the lone pairs of PPi^{4-} , which causes a contraction of the bond length of the P=O bond. The P=O bond length in PPi^{4-} is 1.55 Å, while the P=O bond length in $\text{Zn}(\text{PPi})_2^{6-}$ is 1.53 Å, as shown in **Supplementary Fig.5** in the revised supporting information.

Action: We have supplemented the sentence in the revised manuscript as follows:

(Page 8 in the revised manuscript)

“It is noteworthy that the bond length of P=O in PPi^{4-} is 1.55 Å, whereas that of P=O in $\text{Zn}(\text{PPi})_2^{6-}$ is 1.53 Å (**Supplementary Fig. 5**), which accounts for the blue-shifting in the infrared absorption after coordination (**Fig. 1d**).³⁰”

The relative reference has been also added in the revised manuscript as follows:

30. Fujita, J.; Nakamoto, K.; Kobayashi, M., Infrared spectra of metallic complexes. I. The effect of coordination on the infrared spectra of ammine, rhodanato and azido complexes. *J. Am. Chem. Soc.* **1956**, 78 (14), 3295-3297.

(Page 6 in the revised supporting information)

Supplementary Fig. 5 (a) The optimized bond length of P=O for PPI^{4-} . (b) The optimized bond length of P=O for $\text{Zn}(\text{PPi})_2^{6-}$.

Question-18: L125: Why did you chose a molar ratio of 3:1 and not 2:1?

Response: Thank you for your good question. In theory, $\text{Zn}(\text{PPi})_2^{6-}$ can be prepared with an accurate molar ratio of 2: 1 ($[\text{PPI}^{4-}] : [\text{Zn}^{2+}]$). However, in practical preparation process, such molar ratio (2:1) cannot ensure all the Zn^{2+} has been converted into the $\text{Zn}(\text{PPi})_2^{6-}$. In fact, the formation of $\text{Zn}(\text{PPi})_2^{6-}$ depends on the replacement of water molecules from $\text{Zn}(\text{H}_2\text{O})_6^{2+}$ by PPI^{4-} in sequential steps. Based on the Le Chatelier-Braun's principle, excess ligands (PPI^{4-}) can promote the formation of metal complexes and improve its stability. Similar strategy (i.e., excess ligands) has been widely used to prepare metal complexes.

Action: We have added the relevant description in the revised manuscript as follows:

(Page 6 in the revised manuscript)

“Based on Le Chatelier-Braun's principle, excess PPI^{4-} ligands promote the formation of the $\text{Zn}(\text{PPi})_2^{6-}$ complex and improve its stability.”

Question-19: L125: How did you determine the solubility?

Response: Thanks for your good question. Since we have prepared a solution in which the concentration of Zn^{2+} and the concentration of PPI^{4-} are fixed, we only need to concentrate the solution at 50°C using a rotary evaporator until a very small amount of crystals appear, at which

point the volume of the saturated solution can be measured by a measuring cylinder. The solubility can be calculated according to the law of conservation of elements.

For example, the $\text{Zn}(\text{PPi})_2^{6-}$ solution with a $[\text{PPi}^{4-}]:[\text{Zn}^{2+}]$ molar ratio of 3:1 was prepared with 40 mL of 1 M ZnBr_2 and 25 mL of 3 M K_4PPi , and then concentrated under reduced pressure to 45 mL at 50 °C. The solubility can be calculated to be $\frac{1\text{M}\times 40\text{ mL}}{45\text{ mL}} = 0.9\text{ M}$. The other solubilities of $\text{Zn}(\text{PPi})_2^{6-}$ solution with different $[\text{PPi}^{4-}]:[\text{Zn}^{2+}]$ molar ratios are calculated in the same method.

Question-20: L138: Why do you compare to ZnBr_2 ? Because of Zn-Br flow batteries? I think it would be good to spend 1-2 sentences explaining this.

Response: Thanks for your good suggestion. Yes, we compare to ZnBr_2 electrolyte because ZnBr_2 electrolyte is usually utilized in zinc-based RFBs, and we also performed the parallel ZIFBs using ZnBr_2 electrolyte and $\text{Zn}(\text{PPi})_2^{6-}$ electrolyte in Figure 4a.

Action: We have added a sentence in the revised manuscript as follows:

(Page 7 in the revised manuscript)

“Here, we chose ZnBr_2 electrolyte for comparison because it is widely used in zinc-based flow batteries.”

Question-21: L148: Is this concentration dependent?

Response: Thanks for your question. The chemical shifts of substances at different concentrations could not change. See *Magn. Reson. Chem.* 2018; 56: 1124-1130.

Question-22: L160: Please explain ESP with 1-2 sentences.

Response: Thanks for your good suggestion. ESP (Electrostatic Potential) of a molecule refers to a physical quantity that describes the properties of the electrostatic field surrounding the molecule, which is usually calculated to understand the molecular reactivity.

Action: We have revised the relevant sentences in the revised manuscript as follows:

(Page 8 in the revised manuscript)

“To understand the charge distribution and electron density of two species, electrostatic potential (ESP) mapped molecular van der Waals surfaces of them were also calculated. The results exhibit

total different electric inherent, i.e., positive ESP for $\text{Zn}(\text{H}_2\text{O})_6^{2+}$ ion, while negative ESP for $\text{Zn}(\text{PPi})_2^{6-}$ ion.”

Question-23: L165: The molecule has a 6- charge but it is electrophilic? How does that work? Why do you reference Fig 2d here?

Response: Thank you for pointing out our wrong description. The $\text{Zn}(\text{PPi})_2^{6-}$ with negative ESP should be nucleophilic, rather than “electrophilic”. In our negolyte, $\text{Zn}(\text{PPi})_2^{6-}$ (with negative ESP) seldom combines with OH^- to form $\text{Zn}(\text{OH})_2$ precipitates. In contrast, $\text{Zn}(\text{H}_2\text{O})_6^{2+}$ (with positive ESP) in conventional electrolytes can react with OH^- to generate undesired $\text{Zn}(\text{OH})_2$ precipitates. Besides, we mistakenly referenced Figure 2d here. In fact, Figure 2h should be referenced here to indicate the negative ESP around $\text{Zn}(\text{PPi})_2^{6-}$ ion.

Action: We have revised the manuscript as follows:

(Page 8 in the revised manuscript)

“In contrast, $\text{Zn}(\text{PPi})_2^{6-}$ with negative ESP (**Fig. 2h**) is nucleophilic, which would contribute to the suppression of the by-product (e.g. $\text{Zn}(\text{OH})_2$ and ZnO) formation during zinc deposition.”

Question-24: L170: Can you elucidate why the LUMO of the complex is higher than the free Zn^{2+} ?

Response: Thank you for your good question. Both $\text{Zn}(\text{PPi})_2^{6-}$ and $\text{Zn}(\text{H}_2\text{O})_6^{2+}$ are coordination complexes where zinc is the central metal ion, but they differ in their ligands. The basicity of the ligand can affect the corresponding LUMO energy level of the complex. For example, Laia Vilella et al. found that the LUMO energy of $[\text{Ir}(\text{O})(\text{H}_2\text{O})(\text{phpy})_2]^+$, $[\text{Ir}(\text{O})(\text{OH})(\text{phpy})_2]$ and $[\text{Ir}(\text{O})_2(\text{phpy})_2]^-$ increased successively with increasing ligand basicity. (*Dalton Trans.*, 2011,40, 11241-11247). Here, PPi^{4-} ions are stronger bases than water molecules. This increased ligand basicity can lead to a stronger interaction with the positive zinc ion, affecting the electronic structure around the metal center. The stronger interaction can result in a higher LUMO energy for $\text{Zn}(\text{PPi})_2^{6-}$ compared to $\text{Zn}(\text{H}_2\text{O})_6^{2+}$.

Action: We have revised the corresponding description in the revised manuscript as follows:

(Page 8 in the revised manuscript)

“The results show that $\text{Zn}(\text{PPi})_2^{6-}$ owns a higher LUMO energy (-0.17 eV) than $\text{Zn}(\text{H}_2\text{O})_6^{2+}$ (-1.62 eV), which is attributed to the stronger ligand basicity of PPi^{4-} ions than water molecules.³⁵”

The relative reference has been also added in the revised manuscript as follows:

35. Vilella, L.; Vidossich, P.; Balcells, D.; Lledós, A., Basic ancillary ligands promote O-O bond formation in iridium-catalyzed water oxidation: A DFT study. *Dalton Trans.* **2011**, *40* (42), 11241-11247.

Question-25: L204: Why is it described as if surprising that Zn^{2+} goes through a cation exchange membrane? What else would you expect?

Response: Thank you for your questions. We would like to emphasize that free Zn^{2+} can permeate through a cation exchange membrane, whereas $Zn(Pi)_2^{6-}$ cannot. As previously mentioned (see response to question 14), Zn^{2+} ions transferring to the catholyte do not undergo reactions in that environment. However, due to the limited speed of Zn^{2+} ion migration back to the negolyte, there is a reduction in the total Zn^{2+} concentration within the negolyte during rapid zinc deposition on the anode. This issue does not arise with $Zn(Pi)_2^{6-}$ negolyte.

Question-26: Figure 4a: There are bumps in the blue curves at ca ., 20 mAh and 90 mAh. What is going on there?

Response: Thank you for your good question. In Figure 4a, we set the depth of charge of $Zn(Pi)_2^{6-}$ to be close to 100%, which means that the concentration of $Zn(Pi)_2^{6-}$ in the compact layer (see **Figure Answer 1.**) is very low after charging, and the coordination equilibrium at the surface of electrode would be broken. When the cell is discharged, the dissolution of Zn^{2+} would recombine with the Pi^{4-} to construct a stable double electronic layer, which results in a decreased discharge voltage in the initial stage. For other rate or long cycle battery tests, we have set depths of charge close to 80%, there are no curve fluctuations observed.

Action: We have added the relevant description in the revised manuscript as follows:

(Page 11 in the revised manuscript)

“The fluctuation in the GCD curves of $Zn(Pi)_2^{6-}$ based ZIFB may be due to the disruption of the coordination equilibrium of $Zn(Pi)_2^{6-}$.”

Question-27: Figure 4f: Why is the charge polarization decreasing upon cycling?

Response: Thank you for your question. The charging polarization is reduced when zinc is deposited.

Since the Coulombic efficiency is slightly below 100%, the residual zinc will be in close contact with the carbon felt, the interfacial resistance of the electrodes is thus reduced, and accordingly the charging voltage plateau of the cell will be decreased after several cycles.

Action: We have added the relevant description as follows:

(Page 12 in the revised manuscript)

“The charging voltage polarization decreases gradually after a few cycles due to the close contact of the residual zinc with the carbon felt.”

Question-28: L224: Again, why is it presented as surprising that the negatively charged chelate does not go through the cation exchange membrane?

Response: Thank you for your question. In theory, negatively charged complexes should be effectively isolated by cation exchange membranes. However, in practical applications, there may be trace amounts of negatively charged complexes that shuttle through the cation exchange membrane (Joule 2019, 3, 1-15). In our case, no detectable $\text{Zn}(\text{PPI})_2^{6-}$ ions were observed after 30 days (**Supplementary Fig. 14**) in the reference cell, suggesting an exceptionally slow permeation rate of $\text{Zn}(\text{PPI})_2^{6-}$ ions through the JCM-D membrane.

Question-29: L292: Coulombic interaction between the membrane and the electrolyte, why would this affect the dendrite formation?

Response: Thank you for your good question. Coulombic interaction between the membrane and the electrolyte can alleviate the Zn-dendrite formation, which has been reported by Li's group in 2018 (Ref. 47: *Nat. Commun.* 2018, 9, 3731). Li *et al.* have demonstrated that $\text{Zn}(\text{OH})_4^{2-}$ negolyte easily formed Zn dendrites upon deposition using non-charged porous PES membrane, while it didn't grow dendrites using porous PES/SPEEK composite membrane with negatively charged groups. The authors claimed that this repulsion between $\text{Zn}(\text{OH})_4^{2-}$ and the PES/SPEEK composite membrane with negatively charged groups can force Zn to be plated toward the carbon felt direction, thus inhibiting the Zn dendrite growth.

Question-30: L294: The high nucleation overpotential indicates that there should be a ca. 200 mV overpotential for plating Zn in the pyrophosphate solution? But you do not observe such a cell

polarization?

Response: Thank you for your good questions. This is the Zn nucleation overpotential (We marked it in the revised supporting information, labeled as **Supplementary Fig. 25**) at the initial stage, but not the real overpotential for the overall charging process. To show this more clearly, we have modified the horizontal coordinate in Figure 4a to start from -1 mAh. The charging polarization for $\text{Zn}(\text{PPi})_2^{6-}$ based ZIFB is still high up to 50 mV at the initial stage, which is larger than 30 mV of the Zn^{2+} based ZIFB, but it decreases rapidly once Zn is deposited. The whole overpotential of the two cell is close to each other, which can be seen from their voltage difference between charging and discharging plateaus in **Fig. 4a**.

Action: We have revised the supporting information as follows:

(Page 26 in the revised supporting information)

Supplementary Fig. 25 Symmetric zinc-based RFBs based on 0.2 M ZnBr_2 (light blue line) and 0.2 M $\text{K}_6\text{Zn}(\text{PPi})_2$ (orange line) negolytes operated at 40 mA cm^{-2} using noncharged filter membrane. The capacity of pre-deposited Zn on the carbon felt at the positive side is 200 mAh.

(Page 10 in the revised manuscript)

Fig. 4a GCD profiles of 0.2 M Zn(PPi)₂⁶⁻ negolyte based ZIFB (light blue line) and 0.2 M ZnBr₂ based ZIFB (orange line) at 40 mA cm⁻² in the first cycle.

Question-31: L323: to analyze not to analysis

Response: Thank you for your good suggestion. We have corrected this word in the revised manuscript.

Question-32: L335: Discussion  Conclusion: How does your work guide future efforts? Please explain better the key learnings and how they can guide future work.

Response: Thank you for your good suggestion.

Action: We have supplemented a sentence in the revised the manuscript as follows:

(Page 15 in the revised manuscript)

“Further efforts should focus on increasing the cell voltage (by utilizing the high-potential electrolytes) or improving the cycling stability under ultra-high deposited zinc areal capacity to achieve further breakthroughs in energy density or long duration energy storage.”

Response to Reviewer#2

Overall Comment: This manuscript presented a very nice work on negolyte development for zinc-iodine flow battery. Author's innovative refreshing chemistry by chelating $K_4P_2O_7$ with Zn^{2+} has produced a negolyte that enabled a high-voltage and dendrite-free zinc-iodine flow battery, performing significantly better than conventional zinc-iodine flow battery in terms of working current density and areal capacity. Their effort and novelty are to be commended, which will have important impact for flow battery technology. I recommend the manuscript to be accepted and published in Nature Communications.

Response: Thank you very much for recognizing our manuscript. We would like to answer your questions separately and revise the manuscript according to your suggestions. All the revisions according to your questions/suggestions are marked in purple in the revised manuscript.

Question-1: For $Zn(PPi)_2^{6-}$ based negolyte, why was $ZnCl_2$ employed, not $ZnBr_2$ (consistent with the results for $Zn(H_2O)_6^{2+}$?

Response: Thank you for your good question. The solubility of $Zn(PPi)_2^{6-}$ electrolyte prepared from $ZnBr_2$ is 0.7 M at room temperature with a $[PPi^{4-}]:[Zn^{2+}]$ ratio of 3:1, which is lower than that of $Zn(PPi)_2^{6-}$ prepared from $ZnCl_2$.

Action: According to your question, we have added the relevant sentence in the revised manuscript as follows:

(Page 6 in the revised manuscript)

“Besides, when $ZnBr_2$ precursor is used, the solubility of the prepared $Zn(PPi)_2^{6-}$ solution will decrease to 0.7 M with a $[PPi^{4-}]:[Zn^{2+}]$ ratio of 3:1.”

Question-2: For the permeability measurement of Zn^{2+} , the right compartment was filled with 20 mL of 0.4 M KCl, while in Supplementary Fig. 11's caption, saturated zincon monosodium salt is added in the reference cell, is the zincon monosodium salt added into the right compartment prior to the permeability measurement?

Response: Thank you for your good question. Yes, the saturated zincon monosodium salt reagent is added to the right-hand cell in advance, and if trace amounts of zinc ions diffuse from the left compartment, the zincon monosodium salt (yellow in colour) will coordinate with the Zn^{2+} and the

solution will appear in red colour.

Action: We have added related information in the revised manuscript as follows:

(Page 17 in the revised manuscript)

“For the permeability measurement of Zn^{2+} , the left compartment of the diffusion cell was filled with 20 mL of 0.2 M $ZnBr_2$, while the right compartment was filled with 20 mL of 0.4 M KCl, and saturated zincon monosodium salt reagent was added to right compartment in advance.”

Question-3: How about the stability of $Zn(PPi)_2^{6-}$ based negolyte at a high temperature?

Response: Thanks for your good question. We prepared the saturated $Zn(PPi)_2^{6-}$ solution by rotary evaporator at 50 °C, and the high-concentration negolyte can be stably operated. Besides, we heated 0.2 M $Zn(PPi)_2^{6-}$ negolyte at 80 °C over 48 h, no precipitation was found.

Action: We revised the related information in the revised manuscript as follows:

(Page 15 in the revised manuscript)

“The resulting chelated $Zn(PPi)_2^{6-}$ solution was stirred continuously until the solution became transparent, and then concentrated to 45 mL under reduced pressure at 50 °C.”

Question-4: In flow batteries tests, why excess posolyte is used?

Response: Thanks for your good question. The KI posolyte may generate solid I_2 during cycling at high state of charge (SOC), leading to its deposition on the carbon felt and potential pipe clogging. However, the excess KI can coordinate with the I_2 , preventing the formation of I_2 -deposits on the carbon felt. In addition, KI itself is slowly oxidized by air and therefore usually needs to be stored away from light. For these two reasons, we add excess posolyte.

Question-5: Why the battery using Zn^{2+} negolyte shown a much low coulombic efficiency (79%)?

Response: Thanks for your good question. As shown in Figure 5a, the flow battery using Zn^{2+} negolyte exhibits a low Coulombic efficiency of 79%. This phenomenon arises from two primary reasons. Firstly, some Zn^{2+} ions diffuse from the negolyte to the posolyte, crossing the cation-ion membrane during the charging process. Secondly, the electrochemical hydrogen evolution reaction (HER) occurs in the negolyte during the charging process. Please refer to our response to question-6 for detailed information.

Question-6: In line 194-202, Page 7, two parallel near neutral ZIFBs, 0.2 M ZnBr₂ negolyte (pH=5.6), and the other with 0.2 M K₆Zn(Pi)₂ negolyte (pH=9.2), were employed. Is this pH difference affecting CE of the battery? The HER potentials of ZnBr₂ negolyte and K₆Zn(Pi)₂ negolyte is suggested to provided.

Response: Thank you for your good suggestion. In our opinion, the low CE of ZnBr₂ based ZIFB should arise from the crossover of Zn²⁺ from the negolyte to the posolyte and the electrochemical hydrogen evolution reaction (HER) occurs in the negolyte during the charging process. In Figure 4a, these ZIFBs were charged with a fixed capacity of 100 mAh, calculated based on the theoretical capacity of the Zn²⁺ ions in the negolyte. For the ZnBr₂ based ZIFB, before the cell was charged at 100% SOC (based on the used ZnBr₂), partial of Zn²⁺ ions had been permeated from the negolyte to the catholyte, and therefore H⁺ will be reduced (i.e., HER) after the deposition of Zn²⁺ ions.

According to your suggestion, we also performed the LSV test of the 0.2 M ZnBr₂ and 0.2 M Zn(Pi)₂⁶⁻ negolyte on 1 cm² carbon paper at a scan rate of 5 mV s⁻¹. The results showed that the HER in the 0.2 M ZnBr₂ negolyte was more severe than that in 0.2 M Zn(Pi)₂⁶⁻.

Action: We have revised the manuscript as follows:

(Page 11 in the revised manuscript)

“Linear sweep voltammetry (LSV) of the two negolytes was then performed, and the result showed that the hydrogen evolution reaction (HER) in the 0.2 M ZnBr₂ negolyte was more severe than that in the 0.2 M Zn(Pi)₂⁶⁻ negolyte (**Supplementary Fig. 8**).”

(Page 9 in the revised supporting information)

Supplementary Fig. 8 LSV profiles of the 0.2 M ZnBr₂ and 0.2 M Zn(PPi)₂⁶⁻ negolytes on carbon paper (1 cm²) at a scan rate of 5 mV s⁻¹.

Question-7: In line 157, Page 6, Zn²⁺ has a stronger interaction with PPI⁴⁻, how about the desolvation process of Zn(PPi)₂⁶⁻ (or is the dissociation energy of Zn(PPi)₂⁶⁻ on anode surface high?)?

Response: Thank you for your good question. The dissociation process of Zn(PPi)₂⁶⁻ means that it decomposes to Zn²⁺ and PPI⁴⁻. We found that the stability constant of Zn(PPi)₂⁶⁻ is $1 \times 10^{11.0}$, and this value is much lower than that of Zn(OH)₄²⁻ ($1 \times 10^{17.6}$). Note that the latter could plate on the carbon felt in a rapid pace. Hence, it is anticipated that the desolvation process of Zn(PPi)₂⁶⁻ is not too high.

REVIEWER COMMENTS

Reviewer #1 (Remarks to the Author):

The authors have done a very good job revising the manuscript and answered many of my questions in their response, for example the zeta potential part is a great addition. The manuscript contains a lot of hard work, suitable experiments and the authors have well explained the low zinc potential induced by very low free-Zn concentration, which in turn is caused by the high stability of the complex. However, some key points are still unclear to me and I require further explanations before I can recommend publication of the proposed mechanism:

The complex has a stability constant of 1011, it is quite stable indeed. Accordingly, there is only 10⁻¹⁰ free Zn²⁺. You suggest that this is the main reason why the system works without dendrites and at the observed low potential. Since only free Zn²⁺ is plated, the complex has to dissociate rapidly to provide fresh Zn²⁺ to be plated. But what is the driving force for this dissociation? If you charge at high currents, a lot of complex has to dissociate very rapidly to provide enough plate-able Zn²⁺, but with such a stability constant this seems unlikely. Why is enough complex falling apart to provide enough Zn²⁺ at high rates?

If it is presumed to be simply equilibrium driven ($\text{ZnPPi}_2 \rightleftharpoons \text{Zn}^{2+} + 2\text{PPi}$, if free Zn²⁺ is consumed and plated the equilibrium shifts to the right side so some of the complex dissociates to provide fresh Zn²⁺) a much less stable complex would be required to deliver enough Zn²⁺ in time to be plated at 200 mA/cm². What makes the complex fall apart so easily even though it has a stability constant of 1011? Can you calculate how many moles of zinc you are plating per second at 200 mA/cm²? This would inform us about how much of the complex has to dissociate in said amount of time. Essentially, this is what you see in cyclic voltammetry, not enough Zn²⁺ is provided at some point resulting in a cathodic current maximum.

During discharge, on the other hand, you state that any Zn²⁺ released from the anode is immediately complexed. At high currents, there is locally a very large amount of Zn²⁺ stripped from the electrode, which according to the description of your synthesis should form insoluble Zn₂PPi. You claim that there is no local over-concentration of Zn²⁺ upon discharge and that all Zn²⁺ is immediately consumed to form ZnPPi₂ but as you describe the synthesis of ZnPPi₂ there is a certain time component. Can you calculate, again, how much zinc is stripped from the anode per second at 200 mA/cm²? Of course, convection in the flow cell helps here. Can you do zinc plating in a static three electrode setup? I would expect the maximum plating rate to be significantly lower.

The high binding energy of PPI to Zn seems to make it a great additive to modulate Zn plating/stripping. However, the proposed mechanism, i.e., complex dissociation at a rate proportional to free Zn²⁺ consumption and vice versa complex formation as soon as Zn²⁺ is stripped does not convince me. Can you please explain if I am misunderstanding something?

Additional comments:

- I think you should include your explanations to my major concerns 2 and 3 (Nernst and suppressed HER) in the SI
- You should state in the text that your calculations (Figure 5f) are based on Zn without any charge
- In Figure 1d you discuss $\nu_s(\text{PO}_3)$ and $\nu_{as}(\text{PO}_3)$ which obviously shift a lot upon complex formation. Is it possible to isolate the P=O vibrations in IR/Raman and compare those to the bond lengths you calculate?
- Answer to question 19: I think this should be included in the SI
- Figure S4: You state that experimental and calculated spectra fit very well, which I think is a bit of a stretch in this case. Where are the discrepancies coming from?
- Figure S6 is missing
- The zincon discussion in your response to reviewer 2 could be included in the SI
- English needs work

Reviewer #2 (Remarks to the Author):

The authors have adequately addressed the comments. I suggest that the paper be accepted for publication in the Journal.

Response to Reviewer#1

The authors have done a very good job revising the manuscript and answered many of my questions in their response, for example the zeta potential part is a great addition. The manuscript contains a lot of hard work, suitable experiments and the authors have well explained the low zinc potential induced by very low free-Zn concentration, which in turn is caused by the high stability of the complex. However, some key points are still unclear to me and I require further explanations before I can recommend publication of the proposed mechanism:

Response: Thank you very much for reviewing our manuscript and giving many constructive comments. We would like to answer your questions separately and revise the manuscript according to your suggestions. All the revisions according to your questions/suggestions are marked in blue in the revised manuscript and revised SI.

Question-1: The complex has a stability constant of 10^{11} , it is quite stable indeed. Accordingly, there is only 10^{-10} free Zn^{2+} . You suggest that this is the main reason why the system works without dendrites and at the observed low potential. Since only free Zn^{2+} is plated, the complex has to dissociate rapidly to provide fresh Zn^{2+} to be plated. But what is the driving force for this dissociation? If you charge at high currents, a lot of complex has to dissociate very rapidly to provide enough plate-able Zn^{2+} , but with such a stability constant this seems unlikely. Why is enough complex falling apart to provide enough Zn^{2+} at high rates?

Response: Thank you very much for your insightful question. Here, we would like to provide the following response:

As you correctly pointed out, a stability constant of 10^{11} indicates that the free Zn^{2+} concentration is very low, at 10^{-10} M, in the equilibrium state, indicating significant stability of the complex under equilibrium condition. However, it should be noted that such a high stability constant does not necessarily imply a low dissociation rate under non-equilibrium condition. In other words, this stability constant cannot fully reflect the dissociation kinetics under non-equilibrium condition. During the Zn-plating process, as free Zn^{2+} is consumed, dissociation occurs rapidly to maintain the free Zn^{2+} concentration at 10^{-10} M. Similarly, during the Zn-stripping process, as the free Zn^{2+} concentration increases, complexation occurs rapidly to ensure the concentration remains at 10^{-10} M. Our results, particularly the rapid charge/discharge of the flow cell

at high rates, confirm this phenomenon. Similar behavior has been observed in other redox flow batteries, such as chelated ZnBr_4^{2-} or Zn(OH)_4^{2-} based zinc-iron flow batteries, which can operate at current densities exceeding 100 mA cm^{-2} (Please refer to *Energy Storage Mater.* 2022, 44, 433-440 and *Nat. Commun.* 2021, 12, 3409).

In summary, a high stability constant indicates a low concentration at equilibrium condition, but does not necessarily imply a low dissociation rate under non-equilibrium conditions. We mentioned this point in the revised manuscript, as per your suggestion (see our below action for details).

Action: We have revised the manuscript as follows:

(Page 8 in the revised manuscript)

The observed high rate (i.e., 200 mA cm^{-2}) is superior to that of most reported Zn-based flow batteries. This phenomenon suggests that dissociation of Zn(PPi)_2^{6-} (or the complexation of Zn^{2+} and PPi^{4-}) occurs at a very rapid rate during the Zn-plating (or Zn-stripping) process. As previously mentioned, the stability constant (10^{11}) of Zn(PPi)_2^{6-} indicates that the free Zn^{2+} concentration is very low, at 10^{-10} M , in the equilibrium state, signifying significant stability of the complex under equilibrium condition. However, it should be noted that such a high stability constant does not necessarily imply a low dissociation rate under non-equilibrium conditions. During the Zn-plating process, as free Zn^{2+} is consumed, dissociation occurs rapidly to maintain the free Zn^{2+} concentration at 10^{-10} M . Similarly, during the Zn-stripping process, as the free Zn^{2+} concentration increases, complexation occurs rapidly to ensure the concentration remains at 10^{-10} M . Furthermore, the similar behavior has been observed in chelated ZnBr_4^{2-} or Zn(OH)_4^{2-} based zinc-iron flow batteries.^{19,47}

(Page 19 in the in the revised manuscript)

47. Hu, J., Tang, X., Dai, Q.; Liu, Z.; Zhang, H.; Zheng, A.; Yuan, Z.; Li, X., Layered double hydroxide membrane with high hydroxide conductivity and ion selectivity for energy storage device. *Nat. Commun.* **2021**, *12*, 3409.

Question-2: If it is presumed to be simply equilibrium driven ($\text{ZnPPi}_2 \leftrightarrow \text{Zn}^{2+} + 2\text{PPi}$, if free Zn^{2+} is consumed and plated the equilibrium shifts to the right side so some of the complex dissociates to provide fresh Zn^{2+}) a much less stable complex would be required to deliver enough Zn^{2+} in time to be plated at 200 mA/cm^2 . What makes the complex fall apart so easily even though it has a stability constant of 10^{11} ? Can you

calculate how many moles of zinc you are plating per second at 200 mA/cm²? This would inform us about how much of the complex has to dissociate in said amount of time. Essentially, this is what you see in cyclic voltammetry, not enough Zn²⁺ is provided at some point resulting in a cathodic current maximum.

Response: Thank you very much for your constructive question. Here, we would like to provide the following response:

As we mentioned above, a high stability constant does not necessarily imply a low dissociation rate under non-equilibrium conditions. For instance, a high stability constant of 10^{17.6} for the Zn(OH)₄²⁻ electrolyte implies a very low concentration of Zn²⁺ ions at equilibrium state, resulting in a negative shifted Zn plating/stripping potential to -1.21 V vs. SHE. However, the Zn(OH)₄²⁻ based zinc-iron flow battery can still be cycled at 200 mA cm⁻² with a high energy efficiency of up to 82.4% (Please refer to Nat. Commun. 2021, **12**, 3409). The above literature and our experimental data can fully illustrate that under the non-equilibrium condition, the Zn²⁺ ions in the complexes can dissociate rapidly and achieve reversible plating/stripping.

Besides, we have calculated the number of moles of plated Zn per second at 200 mA cm⁻² on the electrode (5 cm²) as follows:

$$\frac{0.2 \text{ A cm}^{-2} \times 5 \text{ cm}^2 \times 1 \text{ s}}{2 \times 96485 \text{ C mol}^{-1}} = \frac{1 \text{ C}}{2 \times 96485 \text{ C mol}^{-1}} = 0.52 \times 10^{-6} \text{ mol}$$

The depth of the battery chamber on the negative electrode side is 0.2 cm, which corresponds to a volume of 1 mL for the electrolyte it can hold. The total amount of Zn(PPi)₂⁶⁻ ions in the battery chamber is 0.8 M × 1 mL = 0.8 × 10⁻³ mol. It means that the plating current density of 200 mA cm⁻² could be achieved as long as 0.65% of the Zn(PPi)₂⁶⁻ electrolyte in the battery chamber dissociates to Zn²⁺ per second.

Action: we have added the related discussion in the SI as follows:

(Page 28 in the revised SI)

The number of moles of plated Zn per second at 200 mA cm⁻² on the electrode (5 cm²) is:

$$\frac{0.2 \text{ A cm}^{-2} \times 5 \text{ cm}^2 \times 1 \text{ s}}{2 \times 96485 \text{ C mol}^{-1}} = \frac{1 \text{ C}}{2 \times 96485 \text{ C mol}^{-1}} = 0.52 \times 10^{-6} \text{ mol}$$

The volume of negolyte in the cell chamber is 0.2 cm × 5 cm² = 1 mL. The total amount of Zn(PPi)₂⁶⁻ ions is 0.8 M × 1 mL = 0.8 × 10⁻³ mol. It means that the plating current density of 200 mA cm⁻² could be achieved as long as 0.65% of the Zn(PPi)₂⁶⁻ electrolyte

in the battery chamber dissociates to Zn^{2+} per second.

Question-3: During discharge, on the other hand, you state that any Zn^{2+} released from the anode is immediately complexed. At high currents, there is locally a very large amount of Zn^{2+} stripped from the electrode, which according to the description of your synthesis should form insoluble Zn_2PPI . You claim that there is no local over-concentration of Zn^{2+} upon discharge and that all Zn^{2+} is immediately consumed to form ZnPPI_2 but as you describe the synthesis of ZnPPI_2 there is a certain time component. Can you calculate, again, how much zinc is stripped from the anode per second at 200 mA/cm^2 ? Of course, convection in the flow cell helps here. Can you do zinc plating in a static three electrode setup? I would expect the maximum plating rate to be significantly lower.

Response: We appreciate your good questions and are pleased to offer the following response:

The number of moles of Zn^{2+} stripped from the electrode (5 cm^2) per second is equal to the number of moles of Zn plated onto the electrode from the negolyte per second. That is $0.52 \times 10^{-6} \text{ mol}$. For the $0.8 \text{ M Zn}(\text{PPI})_2^{6-} + 0.8 \text{ M PPI}^{4-}$ negolyte, based on the 78% state of charge of the battery, the total amount of PPI^{4-} ions in the battery chamber after the charging process can be calculated as follows:

$$(0.8 \text{ M} \times 2 \times 78\% + 0.8 \text{ M}) \times 1 \text{ mL} = 2.05 \times 10^{-3} \text{ mol}$$

Correspondingly, the molar ratio of stripped Zn^{2+} with the PPI^{4-} is $\frac{0.52 \times 10^{-6} \text{ mol}}{2.05 \times 10^{-3} \text{ mol}} = 0.25\%$. The metal-to ligand molar ratio is quite low, which can prevent the formation of Zn_2PPI side product.

In addition, we performed the zinc plating in a static three-electrode setup, and the result is shown in the revised **Supplementary Fig. S27**. It was found that the initial plating current density exceeded 140 mA cm^{-2} when the polarization potential was controlled at -1.5 V vs. Ag/AgCl (i.e. -1.29 V vs. SHE). In fact, the plating current density decreased rapidly with time in the static three-electrode test, which suggests that the rate of diffusion of $\text{Zn}(\text{PPI})_2^{6-}$ ions to the electrode surface determines the plating rate.

Action: The related sentences are now added in the revised manuscript as follows:

(Page 9 in the revised manuscript)

In addition, to demonstrate the ability of the $\text{Zn}(\text{PPI})_2^{6-}$ electrolyte to be plated at high

rate. We also performed a static three-electrode test (**Supplementary Fig. 20**), and it was found that the initial plating current density exceeded 140 mA cm^{-2} when the polarization potential was controlled at $-1.5 \text{ V vs. Ag/AgCl}$ (i.e. -1.29 V vs. SHE), indicating a high dissociation kinetic of Zn(PPi)_2^{6-} . The rapid decrease in plating current density with time suggests that the rate of diffusion of Zn(PPi)_2^{6-} ions to the electrode surface determines the plating rate.

(Page 21 in the revised SI)

Supplementary Fig. 20 Zinc plating profile in $0.8 \text{ M Zn(PPi)}_2^{6-}$ electrolyte using the constant potential polarization method. Carbon felt (1 cm^2) was served as the working electrode, zinc foil (0.2 mm thick) as the counter electrode and Ag/AgCl as the reference electrode, respectively. The plating potential was set at $-1.5 \text{ V vs. Ag/AgCl}$.

Question-4: The high binding energy of PPI to Zn seems to make it a great additive to modulate Zn plating/stripping. However, the proposed mechanism, i.e., complex dissociation at a rate proportional to free Zn^{2+} consumption and vice versa complex formation as soon as Zn^{2+} is stripped does not convince me. Can you please explain if I am misunderstanding something?

Response: Thanks for your good question. According to the binding energy of PPI^{4-} to Zn^{2+} , the concentration of free Zn^{2+} ions is low at the equilibrium state. However, this does not mean that the dissociation rate of Zn(PPi)_2^{6-} is slow when the equilibrium is broken. In other words, when the coordination equilibrium is broken (e.g., when Zn^{2+} is consumed), the complex can dissociate and undergo electrochemical deposition at a rapid rate to maintain the free Zn^{2+} concentration at 10^{-10} M . In short, a low concentration at equilibrium does not mean that the dissociation rate is also slow. This

has been well demonstrated by our data.

Action: We have revised the manuscript as follows:

(Page 8 in the revised manuscript)

However, it should be noted that such a high stability constant does not necessarily imply a low dissociation rate under non-equilibrium conditions. During the Zn-plating process, as free Zn^{2+} is consumed, dissociation occurs rapidly to maintain the free Zn^{2+} concentration at 10^{-10} M. Similarly, during the Zn-stripping process, as the free Zn^{2+} concentration increases, complexation occurs rapidly to ensure the concentration remains at 10^{-10} M.

Additional comments:

Question-5: I think you should include your explanations to my major concerns 2 and 3 (Nernst and suppressed HER) in the SI.

Response: Thanks for your valuable suggestion. We have added our explanation of Nernst equation and the suppression of HER in the revised SI.

Action: We have revised the manuscript as follows:

(Page 3 in the revised SI)

For the 0.1 M $Zn(PPi)_2^{6-}$ negolyte (0.1 M $ZnCl_2$ + 0.3 M K_4PPi) used for CV test in **Figure 3a**, the concentration of Zn^{2+} at the equilibrium state was set as “x”, the concentration of $Zn(PPi)_2^{6-}$ was 0.1-x, and the concentration of PPi^{4-} was 0.1+2x.

Initial state	0.1	0.3	0
Equilibrium state	x	0.1+2x	0.1-x

According to the stability constant of $Zn(PPi)_2^{6-}$ (Log K = 11.0), the equation is given as follows:

$$\frac{0.1 - x}{x(0.1 + 2x)^2} = 10^{11}$$

The x-value is estimated to be 1×10^{-10} . It means that the concentration of Zn^{2+} in 0.1 M $Zn(PPi)_2^{6-}$ negolyte is 1×10^{-10} M. Considering the Nernst Equation:

$$\varphi_{Zn(PPi)_2^{6-}/Zn} = \varphi_{Zn^{2+}/Zn} + \frac{RT}{nF} \ln[Zn^{2+}] = -0.76 \text{ V} + 0.029 \log[Zn^{2+}]$$

It is believed such a low concentration (1×10^{-10} M) of Zn^{2+} well explains the negative

shift of the Zn plating/stripping potential in the CV tests.

(Page 28 in the in the revised SI)

In conventional aqueous electrolytes containing large amounts of free Zn^{2+} , the formation of hydrated Zn^{2+} is inevitable. During the Zn plating process, H_2O molecules in the solvated structure of Zn^{2+} can gain electrons, resulting in HER. Such undesired HER not only reduces the Coulombic efficiency, but also leads to inhomogeneous Zn plating. In the subsequent Zn plating, the top effect of these inhomogeneous Zn deposits can aggravate the dendrite growth. In the $\text{Zn}(\text{PPi})_2^{6-}$ electrolyte (e.g., 0.1 M ZnCl_2 + 0.3 M K_4PPi), the concentration of free Zn^{2+} is very low ($\sim 10^{-10}$ M), and therefore H_2O molecules remain in the free solvent network or in conjunction with K^+ (1.2 M),^{S1} rather than generating hydrated Zn^{2+} . As a result, the $\text{Zn}(\text{PPi})_2^{6-}$ electrolyte mitigates the undesired HER and promotes smooth Zn plating.

(Page 31 in the in the revised SI)

S1. Yang, C., Xia, J., Cui, C., Pollard, T.P., Vatamanu, J., Faraone, A., Dura, J.A., Tyagi, M., Kattan, A., Thimsen, E., Xu, J.; Song, W.; Hu, E.; Ji, X.; Hou, S.; Zhang, X.; Ding, M. S.; Hwang, S.; Su, D.; Ren, Y.; Yang, X.; Wang, H.; Borodin, O.; Wang, C. All-temperature zinc batteries with high-entropy aqueous electrolyte. *Nat. Sustain.* **2023**, *6*, 325-335.

Question-6: You should state in the text that your calculations (Figure 5f) are based on Zn without any charge.

Response: Thank you for your good suggestion. We have added related sentence in revised the manuscript as follows:

Action: We have revised the manuscript as follows:

(Page 11 in the revised manuscript)

It is found that the binding energy between a PPi^{4-} ion and the uncharged Zn(101) surface is -1.05 eV (**Fig. 5f**), which is significantly higher than that between a H_2O molecule and the uncharged Zn(101) surface (-0.32 eV).

Question-7: In Figure 1d you discuss $\nu_s(\text{PO}_3)$ and $\nu_{as}(\text{PO}_3)$ which obviously shift a lot upon complex formation. Is it possible to isolate the P=O vibrations in IR/Raman and compare those to the bond lengths you calculate?

Response: Thanks for your good suggestion. We consider it inappropriate to isolate the

P=O vibration from the P-O vibration in $\text{Zn}(\text{PPi})_2^{6-}$, since the entire PO_3 group in $\text{Zn}(\text{PPi})_2^{6-}$ is in a large conjugated system. The IR vibration peak of the unconjugated P=O double bond is usually located at 1275 cm^{-1} (Adv. Mater. Interfaces 2018, 5, 1800423). However, no obvious IR vibration peaks were found between $1200\text{-}1300\text{ cm}^{-1}$ for $\text{Zn}(\text{PPi})_2^{6-}$.

Question-8: Answer to question 19: I think this should be included in the SI.

Response: Thank you for your valuable suggestion. We have included the solubility test and Supplementary Fig. 3 in the revised SI.

Action: We revised the manuscript as follows:

(Page 4 in the revised SI)

Supplementary Fig. 3 The solubility and conductivity of $\text{Zn}(\text{PPi})_2^{6-}$ at different concentration ratio of PPI^{4-} and Zn^{2+} .

Solubility test: The $\text{Zn}(\text{PPi})_2^{6-}$ solution with a $[\text{PPI}^{4-}]:[\text{Zn}^{2+}]$ molar ratio of 3:1 was prepared with 40 mL of 1 M ZnBr_2 and 25 mL of 3 M K_4PPi , and then concentrated to 45 mL at $50\text{ }^\circ\text{C}$ under reduced pressure. The solubility is thus calculated to be $\frac{1\text{M}\times 40\text{ mL}}{45\text{ mL}} = 0.9\text{ M}$. The other solubilities of $\text{Zn}(\text{PPi})_2^{6-}$ solutions with different $[\text{PPI}^{4-}]:[\text{Zn}^{2+}]$ molar ratios can be calculated by the same method.

Question-9: Figure S4: You state that experimental and calculated spectra fit very well, which I think is a bit of a stretch in this case. Where are the discrepancies coming from?

Response: Thank you for your good question. The theoretically calculated frequencies

are the resonant frequencies and the non-resonant effects are ignored, so the calculated frequencies usually deviate from the actual values.

Action: We revised the manuscript as follows:

(Page 6 in the revised manuscript)

The optimized structure of $\text{Zn}(\text{PPi})_2^{6-}$ is shown in **Fig. 2f**, and the simulated infrared spectrum of $\text{Zn}(\text{PPi})_2^{6-}$ is relatively close to the experimental spectrum (**Supplementary Fig. 4**).

Question-10: Figure S6 is missing.

Response: Thank you for your suggestion. We have added **Supplementary Fig. 6** in the revised SI.

(Page 7 in the revised SI)

Supplementary Fig. 6 Tafel plots for Zn plating/stripping in 0.2 M ZnBr_2 solution at a scan rate of 0.1 mV s^{-1} .

Question-11: The zincon discussion in your response to reviewer 2 could be included in the SI.

Response: Thanks for your insightful suggestion. We have added the related discussion after **Supplementary Fig. 13** in the revised SI.

(Page 14 in the revised SI)

The saturated zincon monosodium salt reagent is added to the right-hand cell in advance, and if trace amounts of zinc ions diffuse from the left compartment, the zincon

monosodium salt (yellow in colour) will coordinate with the Zn^{2+} and the solution will appear in red colour.

Question-12: English needs work.

Response: Thanks for your valuable suggestion. We have try our best to revise the expression in the revised manuscript.

Response to Reviewer#2

The authors have adequately addressed the comments. I suggest that the paper be accepted for publication in the Journal.

Response: Thanks for your very much!

REVIEWERS' COMMENTS

Reviewer #1 (Remarks to the Author):

The authors have done an outstanding job revising the manuscript. I commend their good work and recommend publication.

Reviewer #2 (Remarks to the Author):

This manuscript can be published.